# On Prayer and Dialectic in Modern Jewish Philosophy: Hermann Cohen and Franz Rosenzweig

Ronen Pinkas 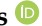

Faculty of Arts, School of Jewish Theology, University of Potsdam, 14469 Potsdam, Germany; pinkas@uni-potsdam.de

**Abstract:** This paper is founded on two philosophical assumptions. The first is that there is a difference between two patterns of recognition: the dialectical and the dialogical. The second assumption is that the origins of the dialogical pattern may be found in the relationship between human beings and God, a relationship in which prayer has a major role. The second assumption leads to the supposition that the emphasis of the dialogic approach on moral responsibility is theologically grounded. In other words, the relationship between humanity and God serves as a paradigm for human relationships. By focusing on Hermann Cohen and Franz Rosenzweig, in the context of prayer and dialectic, this paper highlights the complexity of these themes in modern Jewish thought. These two important philosophers utilize dialectical reasoning while also criticizing it and offering an alternative. The conclusions of their thought, in general, and their position on prayer, in particular, demonstrate a preference for a relational way of thinking over a dialectical one, but without renouncing the latter.

**Keywords:** dialectic; dialogue; prayer; modern Jewish philosophy; religious existentialism

## 1. Dialectic

Since the early Enlightenment, especially the very beginning of the 19th century with the appearance of Hegel's and Fichte's philosophy, the dialectic has earned the title of "the language of philosophy" and the standard bearer of modern values (Gadamer 1980, p. 11; Nikulin 2010).[1] Hegel writes "Kant brought back to memory the dialectic and reinstated it in its position of honor. He did this by elaborating the so-called antinomies of reason [...] Everything that surrounds us can be viewed as an example of the dialectic. [...] The dialectic also establishes itself in all the particular domains and formations of the natural and the spiritual world [...] It is the same principle that forms the basis of all other processes in nature and through which nature is at the same time driven beyond itself" (Hegel 2010, pp. 130–31). It was considered a rational tool that liberated the human being from being subjected to tradition and mythologies based on faith, which contradict logic, yet without completely erasing the early stages. Rather, these stages are integrated into a unified view. Progress was thought to be achieved through the principles of negation, sublation (Aufhebung), and transition to an encompassing higher stage. The dialectical approach—and the assumption that contrast, contradiction, conflict, and paradox are necessary in order to present a picture in its entirety—appears in Hegel's philosophy as a continuous process for arriving at truth, as well as the explanation for the development of spirit and matter. To a considerable extent, modern values were seen as the product of dialectical reasoning. For example, a dialectical negation of the particular individual "self" sublates itself and changes into a "universal I" in the same way that heteronomy sublates itself into autonomy, monarchy into democracy, and tradition to modernity. Hegel's philosophy dialectically placed the philosophical concept above the religious symbol and replaced the theological redemption with the liberation of the absolute spirit. Indeed, from the beginnings of modernity,[2] the processes of secularization, and the growth of empirical and

naturalistic sciences, the trend among significant scholars was the belief that religions and religious worship had become outdated.[3] Various attempts have been made among Jewish thinkers to confront "the crisis of prayer" (Heschel 1998, p. 54)[4] and to revive the status of religious ideas.

In the history of ideas, and despite serious criticisms,[5] dialectic, if using a popularly accepted formulation, succeeded in bringing about a clear distinction between religion, art, ethics, and science.[6] All these and more were seen as the achievements of dialectical reasoning, which was indeed accepted by early modern Jewish thinkers, who tried to emphasize the commonality between the values of Judaism and the values of general culture, in addition to their understanding of Judaism itself in terms of modern philosophy and in the light of scientific progress in general.

## 2. Dialectic and Judaism

Some modern Jewish scholars emphasized the centrality of the dialectic in Judaism, and there is indeed much material that justifies such a consideration. There is an immanent dialectical tension between biblical and Rabbinic Judaism, written and oral Torah, Zionism and diaspora Judaism (the Jerusalem and Babylonian Talmuds), tradition and modernity, reason and faith, and Hebrew as a holy language and everyday language, among other examples. It is not surprising that some have gone so far as to characterize Judaism as a "dialectical religion" (Chamiel 2020, pp. 200–3).[7] The Talmudic style of "pilpul" is often regarded as dialectical reasoning (Boyarin 2017, pp. 47–65). Abraham Heschel writes "Jewish thinking and living can only be adequately understood in terms of a dialectic pattern, containing opposite or contrasting properties" (Heschel 1966, p. 341).[8] It should be noted that Heschel's student, Jacob Neusner, a well-known scholar of rabbinic literature, moderated his teacher's dialectical position. Neusner does not claim that Judaism is dialectical in its essence, but emphasizes in his extensive studies that the sages of the Talmud indeed use a dialectical analysis in their arguments (Neusner 1995, 2005; See also Boyarin 1993, pp. 61–76).

Heschel is unique among modern thinkers in his unequivocal claim that it is not possible to understand Jewish life without adopting a dialectical perspective. According to Heschel, the truth is twofold; therefore, contradictions should be contained in a dialectical synthesis. He brings a Hasidic interpretation to the following Talmudic passage: "the Holy One, praise to Him, seems to be far away when there is no one closer than Him [Jerusalem Talmud, Berachot 9:1]. When we think He is close, then He is remote; when we think He is remote, then He is near (the Baal Shem). The bridge to God is awe" (Heschel 1966, p. 160, with slight changes in translation).[9] In his approach, the complete unitary truth can only be seen from the divine point of view, and is indeed beyond the reach of reason alone. Heschel's "depths theology" (see, for example, Merkle 1985; Kaplan 2007; Giannini 2009) is about hearing God's voice directly, which is beyond language and the mental process of conceptualizing and the formation of symbols.[10]

Heschel's attitude to prayer similarly includes some use of dialectic, expressly in his discussions of "the polarity of prayer" (Heschel 1954, pp. 64–66, 100–2). According to Heschel, prayer is not the initiative of the human being, rather his response to the divine questioning. "Prayer is not a need but an ontological necessity" that "constitutes the very essence of man" (Heschel 1998, p. 78; See also Horwitz 1999). According to Heschel, there are different poles to prayer, which generate contradictions: God and the human being; spontaneity and continuity; prayer and life. Only when these contradictions are considered together in a dialectical synthesis is it possible to restore unity. He writes "Since each of the two [order and outburst, regularity and spontaneity, uniformity and individuality, law and freedom, empathy and self-expression, insight and sensitivity, creed and faith, the word and that which is beyond words] moves in the opposite direction, equilibrium can only be maintained if both are of equal force" (Heschel 1998, pp. 64–65).[11] Heschel sees prayer as an ontological necessity,[12] as a means of becoming aware of the divine presence,

and being known by Him. For Heschel, fulfilling Jewish law (mitzvot) and prayer are the ways to hear the divine voice. Heschel aspired to establish a direct relationship between the human being and God. However, for him, prayer is not a dialogue. He writes "I am not ready to accept the ancient concept of prayer as dialogue. The better metaphor would be to describe prayer as an act of immersion, comparable to the ancient Hebrew custom of immersing oneself completely in the waters as a way of self-purification to be done over and over again" (In Kaplan 1996, p. 179, footnote 20). Prayer according to Heschel is closer to an act of worship, pilgrimage, and sacrifice than to a dialogue (Heschel 1954, p. 33). He writes "Prayer is not a substitute for sacrifice. Prayer is sacrifice. What has changed is the substance of sacrifice: the self took the place of the thing. The spirit is the same. [...] We do not sacrifice. We are the sacrifice" (Heschel 1954, p. 71). Kaplan argues that Heschel's extreme proclamations, such as "Prayer is of no importance unless it is of supreme importance" and "If God is unable to listen to us, then we are insane in talking to Him" (Kaplan 1996, p. 151), demonstrate Heschel's attempt to maintain *the absolute* through the notions of polarity and a dynamic coexistence of contraries. In the context of the dialectic, Heschel recruits a dialectical logic in order to clarify the complexity of prayer. Kaplan argues that Heschel used dialectic to induce people "to seek holiness" (Kaplan 1996, p. 14). For Heschel, the dialectic is part of the religious experience itself because God's language is always ineffable (Kaplan 1996, pp. 42–43, 69). That is, the paradox of the near and far God is infinite. In Heschel's perspective, God is absolute and transcends all synthesis. Hence, prayer is a noble and necessary act of worshiping the absolute, but worship does not entail engaging in a dialogue.

## 3. An Alternative to Dialectic?

Admittedly, dialectical reasoning has been prevalent in the thought of Jewish philosophers from the beginning of modernity to the present. However, concurrently, a critical position developed in relation to various aspects of the Hegelian dialectic. Hermann Cohen and especially Franz Rosenzweig (which is further discussed in detail) belong to the philosophical stream (as do Martin Buber, Emmanuel Levinas, and, in our time, Ephraim Meir) that responds to and negates elements in Hegel's philosophy, including the absolutism of reason (or any form of philosophical or religious absolutism), the dialectical monism of the Spirit, Hegel's pantheism and his idolization of the State, as well as his dialectical progress of history (Pöggeler 1984).[13] Despite some differences between Cohen and Rosenzweig, especially regarding the latter, scholars agree that both demonstrate criticism of the philosophical historicism that appeared after Hegel, which also characterized the *Wissenschaft des Judentums* movement of the 19th century (Schweid 2002, pp. 24–25; Meyer 1995, pp. 67–73; Rotenstreich 1973, p. 58; Chamiel 2019, pp. 542–43. On Hegel's approach to Judaism, see Yovel 1996, p. 23).

In contrast to the Hegelian dialectical way of thinking—which is often described as empty abstraction, which logically develops from a negation of the nothing, and eventually strives to reach "the unity of the determinations in their opposition" (Sayers 2022; See also Hegel 2010, pp. 125–33)—several modern Jewish philosophies express a preference for a "relational way of thinking" (my use of this term follows E. Meir 2022). Steven Kepnes argues that modern Jewish philosophers from Mendelssohn and Cohen to Buber and Rosenzweig have championed the power and value of dialogue and relation. He believes that Jewish philosophy's preference for the dialogic, ultimately, might be seen to originate in the biblical notion of "covenant" (Brit) and in the Talmudic notion of "Talmud Torah" (the commandment of Torah study) and the dialogical communal text study (Kepnes 2004, p. 189). In other words, Kaplan believes that there is a theological foundation (i.e., human–God relations) in Jewish philosophical positions that promote relational thinking and dialogue. This paper supports this assumption and concentrates on prayer as the origin of modern humanistic conceptions of dialogue.

Martin Buber's (1878–1965) philosophy is among the most explicit embodiment of a dialogical approach that positions itself as opposed to dialectic.[14] Buber places relations

(encounter and dialogue) not only as the basis for ethics in general, but primarily as the essential foundation of existence itself. This position is philosophically formulated in his *I and Thou* (published in 1923) in the expression, "In the beginning is the relation" (Buber 1970, p. 69),[15] which is a paraphrase of Genesis 1:1. Thus, *relation* and the *dialogical situation* are not only an epistemological phenomenon but an ontological reality. Accordingly, dialogue is not merely a methodical alternative to philosophical analysis; rather, dialogue is an overcoming of the shortcomings of any other (logical-dialectical) reasoning. In 1928, he wrote

> "Judaism regards speech as an event which grasps beyond the existence of mankind and the world. In contradiction to the static of the idea of Logos, the Word appears here in its complete dynamic as 'that which happens.' God's act of creation is speech, but the same is true of each lived moment. [...] Thus, the whole history of the world, the hidden, real world history, is a dialogue between God and his creature, a dialogue in which man is a true, legitimate partner, who is entitled and empowered to speak his own independent word out of his own being. [...] It is only when reality is turned into logic, and A and non-A dare no longer dwell together, that we get determinism and indeterminism, a doctrine of predestination and a doctrine of freedom, each excluding the other. According to the logical conception of truth, only one of two contraries can be true. [In contrast...] The unity of the contraries is the mystery at the innermost core of the dialogue". (Buber 1963, pp. 255–57)

According to Buber, the synthesis of the dialectical movement offers a theoretical-rational solution to the basic paradoxes of life with a monological formula. In this approach, the "unity" it offers (by means of dialectical synthesis) is a kind of *gnosis*, a unique mysterious knowledge,[16] which supposedly solves dualistic problems, but denies the complexity of reality itself. "Gnosis", writes Buber, misunderstands the "meeting" (Buber 1963, p. 262). Unlike this, the dialogical situation sees paradoxes as essential to reality; thus, unity is not a synthesis of the opposite contradicting sides (which forms the paradox) but a continuous "ever changing" meeting and dialogue between them. Buber's reference to "the static idea of the logos", which he defines as an "erroneous idea" in the understanding of the relationship between God, the world, and the human being, echoes Hermann Cohen's criticism of logos (Cohen 1995, p. 48).[17] According to Buber, unity is fragile, because it is a continues *dialogue* between different and separated parts, and not a fixed monological-dialectical synthesis. The unity of the contraries is "the mystery at the innermost core of the dialogue" and not, as one might suppose, the core of dialectic. On the one hand, Buber is aware that dialectical reasoning since the Enlightenment has liberated human beings from the shackles of religious dogmas and forms of established religion.[18] He applied dialectical thought to describe the basic polarities in human existence (determinism vs. indeterminism, predestination vs. freedom), which are characterized by discomfort (aporia) and conflict. On the other hand, he does not accept the dialectical synthesis, which holds that tension is resolved by a transformation of the contradiction to a higher level of abstraction. For him, dialectical relations eventually lead to a non-satisfying monological synthesis of absolute unity in which "reality is turned into logic", which is a sort of *uniformity* (of reason) but not *unity*.[19] Contrarily, dialogical relationships are based on the ontology of affinities, which does not offer a solution to the ongoing contradictions but rather accepts them as "the mystery at the innermost core of the dialogue". He is aware that dialectic in itself is considered a reasonable tool that seemingly comes to replace revelation. Indeed, the dialogical encounter between contradicting opposites is very difficult to actualize. Therefore, Buber claims that Judaism experienced the dialogical unity of contradiction as theophany (Buber 1963, p. 264; See also Buber 2002, pp. 172–76). That is, dialogue is the redemptive power of revelation.

For Buber, a possible dialogue with God is merely a dialogue with the *Eternal Thou*. He writes "In every You we address the Eternal You" (Buber 1970, p. 57). Buber liberates God

from religion by focusing on the religiousness of the *encounter*.[20] This possibility exists in the here and now of every situation in life, and must not be limited to a certain religious or liturgical moment. Hence, a redemptive action in the world—sanctification, the promotion of unity, "longing to establish a living communion with the unconditioned", and "God's realization through man" (Buber 1972, pp. 79–94)[21]—is not dependent on the performance of a religious ritual in certain moments.[22] On the contrary, all these are daily and continuous demands that are part of ordinary life in the present moment. These are fulfilled only by establishing I–Thou relations. In other words, authentic I–Thou relationships embody within themselves the wonder of creation and are revelation itself. That is, if indeed the absolute is revealed through dialogue with the Eternal Thou, then the accomplishment of such a dialogue, or at least the aspiration for it, can be seen in itself as prayer. Buber, however, does not announce this explicitly.[23] In light of this, it is legitimate to raise the question of whether the modern philosophical view has led to a reinterpretation of the traditional liturgical prayer, re-revealed its original meaning, or deviated from it. Indeed, not every form of prayer is a dialogue, and not every genuine dialogue is prayer. Prayer, as Heschel argues, includes within it the tension between the spontaneous and the fixed, and between intention and action. Prayer in the liturgical scriptures is conducted according to laws: how to pray, when to pray, and what to pray. There are fixed times, fixed ways, and fixed texts (Heschel 1998, pp. 64–65). It is not surprising that in this context, a dialectical approach is sometimes used to describe the relationship between the human being and the distant and near God, and to analyze the historical development of prayer from sacrificial rituals, as described later.

Following this, I argue that the dialogical relational approach should be seen as an approach whose genealogy is based on prayer. That is, the understanding of prayer as a dialogic relationship with God eventually leads to dialogic-humanistic approaches, which do not necessarily confirm a theological foundation. Admittedly, a contemporary dialogic approach does not necessarily require a theistic foundation. Nonetheless, I find it valuable to follow the development from the theological to the philosophical and vice versa.

## 4. Prayer as Dialogue: The Origins of a Dialogical Approach

The idea that prayer is essentially a dialogue with God is neither modern nor obvious.[24] The call for God in the Hebrew Bible—for example, "have mercy on me and hear my prayer" (Psalms 4:2)—is commonly understood as longing for God but not as straight dialogue. Even when the Psalmist declares that God can hear and accepts the prayer (Psalms 6:10), this does not necessarily imply a two-sided conversation. Nevertheless, this idea is present in the sources and received an explicit formulation in rabbinic literature:

> "Just like a man whispering into the ear of his friend and the latter understands.
> Can you have a God who is closer than that to His creatures, from mouth to ear?"
> (Jerusalem Talmud, Berakhot 9:1)

The philosophers who have developed this idea (as well as other comparable expressions in the scriptures, see Rosenberg 1996, pp. 69–107) comprehend prayer not only as a ritual, liturgical activity and an epistemological movement within the human being, but also as a dialogue between the human being and God, which is essentially a model for a broader relational approach. The intimate God who understands, loves, atones, and responds to the human being is perceived as the etiological ground for responsiveness and empathy within humanistic relations. Prayer, as the opening of the heart to *something* that is essentially different and transcendent (which cannot and should not be reduced to self-discourse), can be seen as the primordial theological prototype for the art of unmediated listening.[25] First, actual listening means sincere acknowledgement of the other (N. Gordon 2004). Second, the listening in itself involves a transformation in the listener.[26] As Mendes-Flohr describes, a genuine dialogue entails risk: the "danger" is that by truly listening to the other, one might be changed, transformed cognitively and existentially (Mendes-Flohr 2015, p. 3). Hence, if the origin of the dialogue is prayer, then the former,

as a continuation of the latter, includes within it the possibility of the transcendent, the absolute otherness, and the obligation and responsibility that are implied in these relations.

In light of this, I devote the next part of this paper to examining the attitude toward prayer in the thought of two influential Jewish philosophers, Hermann Cohen and Franz Rosenzweig. Both pay careful attention to prayer in their thoughts, and for both, the liberation and completion of the human being are not conditioned by the practice of the logic of pure reflection, dialectical processes of self-awareness, etc. Rather, this is possible through a relational way of thinking, which obtains its meaning and vitality in the encounter with the other.

## 5. Hermann Cohen's Religion of Reason

Hermann Cohen (1842–1918) is known as a neo-Kantian Jewish philosopher who returned to Kant's philosophy to correct the wrong secular philosophies that developed from Hegel's thought, such as they appear, for example, in the historical materialism of Marx and in Nietzsche's extreme individualism, both of which were popular at the time. Cohen revives the a priori position of reason, through which he seeks to understand and analyze religion: "reason is meant to make religion independent of the descriptions supplied by the history of religion. [...] history in itself does not determine the concept of reason" (Cohen 1995, pp. 2–3). This is, of course, an explicit negation of the Hegelian historical approach. In the context of Kantian ethics, Cohen's most explicit expression of the relational way of thinking is expressed in the term *correlation*. This concept refers to the formation of the moral subject on the basis of reciprocal relationships (between the human being and God, and between the I and the other), and not on the basis of dialectical abstractions and synthesis. As we shall see, Cohen's approach to prayer involves both of those issues.

Cohen is among the first modern Jewish philosophers to offer a systematic discussion of prayer within a philosophical framework. In his last work, *Religion of Reason Out of the Sources of Judaism* (published 1919), which some scholars consider to be the fourth and final part of his system of philosophy, Cohen bestows religion a separate place alongside the other main three pillars of philosophy: logic, ethics, and aesthetics. The uniqueness of religion, or why it is almost impossible to exhaust the discussion of religion within the field of ethics, is rooted in the importance and contribution of religion to moral philosophy—mainly, the "discovery of the Thou", which appears in Cohen's criticism of the shortcomings of ethics, especially in the context of sin, atonement, and the renewal of moral consciousness.[27] According to Cohen, the entire system of Halacha (Jewish law) is an expression of morality and should be measured only with regard to this goal. He writes, "Religion itself is moral teaching or it is not religion" (Cohen 1995, p. 33).

Cohen's religious thought is a worthy example in Jewish philosophy of a rational attempt to present a unified view of Judaism as a religion of reason following Kantian ethics. Kant's philosophy won sympathy among Jewish philosophers in the 19th and early 20th centuries, which saw it as enabling a defense of ideas, moral faith, and the spiritual life of idealistic ethics against ethical and secular materialism (see Nahme 2019, especially Chapter 2; Poma 2006, p. 127). In Hegel's approach, the objects of religion and their historical appearances are included within the manifestations of philosophical reason, whereas for Kant, religion is beyond the limits of reason and is not necessarily incorporated into philosophy. In this respect, philosophy is not necessarily a progressive substitute for religion. Cohen's philosophy demonstrates a commitment to Kant's ideas, even in those topics where he recognized the necessity to critique, alter, or differ (Kohler 2018). Generally, Cohen, like Kant, maintains that a human being completes themselves when they rationally fulfill their moral essence. Unlike Kant, Cohen maintains that the human being is not only a universal agent of moral reason but rather a unique individual and, therefore, a correlative being. That is, the realization of the moral essence is not possible only from the evaluation of the relation between pure and practical reason, but from actual relationships with another being.

### 6. Cohen's and Kant's Position on Prayer

A significant difference between Cohen and Kant's approach to religion is revealed in their position on prayer. Unlike Kant, who generally undervalued the particular historical form of prayer, Cohen emphasizes its central importance not only with regard to religion but with regard to morality in general.[28] Cohen's discussion of prayer involves an analysis of the Hebrew bible—mostly Psalms and the writings of the prophets—as well as rabbinic literature. For him, prayer is a religious action that demonstrates practical moral reason. Prayer, according to Cohen, is the "original form of monotheism" (Cohen 1995, p. 371),[29] which establishes the connection between religious knowledge and religious action, and between religion and morality in general. It is an example of how moral monotheism manifests itself concretely in the individual's life as a psychological force in the fulfillment of a moral society. In *Religion of Reason*, Cohen considers prayer as the culmination of Jewish religious law, thereby giving it the status of a super-commandment that encompasses the meaning of the law and how it shapes Jewish existence.

The "entire content of the worship of God" lies in prayer. This total content, in turn, is distilled in the Sh'ma Yisra'el of the daily liturgy: "Hear, O Israel: the Lord our God, the Lord is one" (Dt 6:4). Speaking and hearing this teaching on God's uniqueness during prayer are the Jewish people's basic duty of obedience (Wiedebach 2022, p. 527).

In Jewish thought, prayer is necessarily seen as a combination of two actions: the deed and the intention (Cohen 1995, p. 393).[30] Intention in prayer is considered "work of the heart". Therefore, it is said that "Prayer without intention is like a body without a soul" (see Rosenberg 1996, p. 91).[31] While prayer as a deed is binding (i.e., heteronomous), intention, contends Cohen, cannot be forced. He writes

"The law comes from God; the duty from man. [. . .] God commands man, and the man in his free will takes upon himself the "Yoke of the law". Even according to Kant's teaching, man is not a volunteer of the moral law, but has to subjugate himself to duty. There is but one yoke: that of laws and the kingdom of God" (Cohen 1995, p. 345).

According to Cohen, prayer expresses both sides of the correlation between the human being and God. He emphasizes that the Jew is obligated to pray for "the Kingdom of Heaven"—that is, for the fulfillment of morality in the future social reality. That is, not only must there not be a contradiction between the religious consciousness of prayer and the moral consciousness (e.g., Isaiah 1:15: "Though you pray at length, I will not listen, your hands are stained with crime"), but also the former consciousness nourishes the latter. For Cohen, this obligation does not violate the autonomy of the individual's pure will.

Generally, Kant's claim that prayer has no importance for moral consciousness was a challenge for Cohen, as well as for other Jewish and Christian religious thinkers. In Kant's view, there is no need for religious ritual action. He considers religious practices meant to appease God as superstitions and illusions. Prayer is thus a form of inner speech, a heartfelt desire that has no place within the limits of reason. In terms of moral religious conviction, Kant contends that the only personal wish that can be part of prayer is the desire to be favored by God. Namely, the meaning of prayer is the worshiper's hope that what he is unable to accomplish through the strength of his moral consciousness will be accomplished with divine assistance. However, Kant argues that this hope violates the individual's autonomous conscience. Grace, he claims, is only bestowed upon those who deserve it and is not conditioned by any religious worship or prayer to receive it (Kant 2009, pp. 215–19; See also Sagi and Statman 1993, pp. 149–50; Levy 1989).[32] Simply put, for Kant, religion is not crucial for ethics and the accomplishment of morality. The human being does not need more than the discovery of his or her rational morality.

Cohen adopted Kant's position that religion, like ethics, should be based on the a priori of reason, autonomy of the moral will, and a universal moral principle. However, Cohen claimed that the human being indeed deserves grace and atonement from God, though only after they have put forth their best efforts due to their moral obligation. Unlike Kant, Cohen holds that action precedes belief (this is anchored in biblical and rabbinic thought, "[first] We will do and [then] we will listen", Exodus 24:7), and that "the idea of God"

precedes human morality (Kohler 2018, p. 204). While Kant focused on the relationship between intention and duty, Cohen, in the context of prayer, adds the term "language" as a vital third component. Namely, he attributed the significance of prayer to practical reason. He writes that prayer is the "activity of language... in which the will becomes active in all the means of thought" (Cohen 1995, p. 399). That is, the language of prayer can be seen as a practical actualization of the a priori transcendental value (Ballan 2010, p. 5).

It is important to remember that in Cohen's thought, the concept of God is entirely distinct from the universe,[33] and God is known through his moral attributes (i.e., creator and atoner). The concept of God as described in Jewish prayer meets these two conditions. An example of this is the use that Cohen makes of the Psalm "The nearness of God is my Good" (Ps. 73:28). The term "nearness" expresses the notion that a unity with God is not possible and not a desirable aspiration (Cohen 1995, p. 163). This aligns with Cohen's harsh criticism of all forms of pantheism.[34] In addition, it expresses the constant aspiration for the transcendent, knowing that the relationship with God does not appear in the world of the senses. That is, the Psalms express an awareness of the superiority of the literary style in connection with God over plastic art that was perceived as idolatry by the prophets (Cohen 1995, pp. 53–58). Cohen argues that this literary style comprehends God as an idea or archetype and not as an image or semblance. We should remember that Cohen, as an idealist, held that "an idea" is more "real" than the reality perceived by the senses.[35] For him, prayer is a linguistic-literary expression of a direct and unmediated correlation between the individual and his God. As such, he was critical of forms of religious worship and worldviews that deviate from being a "religion of reason", including the beliefs that salvation depends on God's grace alone and is independent of human moral deeds, and that there is a need for an intermediary between the individual and God (whether through the mediation of material worship, a particular person considered a demi-god, or a metaphysical factor such as the Logos). Additionally, he contends that the anthropomorphic expressions of prayer must be constantly checked and criticized (Kohler 2018).

As for God being known through his moral attributes, Cohen brings a passage from the daily prayer: "In His goodness He constantly renews in each day the work of the beginning" (Cohen 1995, p. 68). This prayer, according to Cohen, underlines the connection between the two separate realms: creation (i.e., logic) and morality (i.e., ethics). God as the creator and forgiver contains two aspects that are related to each other. The rabbinic prayer replaces the biblical idea of creation with the two terms "renewal" and "goodness". This implies that the great miracle is not creation, but rather the continuation of the becoming, the "permanency in change", and that "each point in becoming is a new beginning" (Cohen 1995, p. 70). This position is a component of the rejection of mythological stories of creation, but even more, it is linked to the idea of atonement (see Zank 2000). Hence, just as creation is seen as an act of God's goodness that is continually being renewed, so is his act of atonement, which guarantees the renewal of the individual's moral consciousness.

In *Religion of Reason*, Cohen gives central importance to sin and repentance. For him, all monotheistic prayer is confessional and its purpose is reconciliation and atonement. The recognition of sin and turning away from the path of sin are not only possible but "this possibility of self-transformation makes the individual an I" (Cohen 1995, p. 193). In fact, without the confession of sin—namely, admitting the weaknesses of the human being (such as social injustice and indifference to the suffering of others)—the human being does not become an autonomous moral being. Since the social moral task is endless, and "each sin is nothing but a step on the way", (Cohen 1995, p. 206) the possibility of eternal forgiveness from God is necessary. "I remain man, and therefore I remain a sinner. I therefore am in constant need of God, as the One who forgives sin" (Cohen 1995, p. 212). The language of prayer expresses hope and trust in the good God "of reconciliation and redemption" (Cohen 1995, p. 372).[36]

God as renewer (of both creation and atonement) is consistent with Cohen's concept of the world as becoming and reason as a concept that does not contradict revelation.[37]

In fact, there is a correlation between the two terms, as Cohen asserts "Revelation is the creation of Reason" (Cohen 1995, p. 72). Cohen broadens the definition of the theological term revelation. Revelation is not only the origin of morality, but also "the continuation of creation" (Cohen 1995, p. 71). Hence, it is the precondition for both human reason (i.e., the laws of logic, the possibility of knowledge, etc.) and moral consciousness (cultural and individual moral aspirations). This notion, claims Cohen, can be found in the main daily prayer, the Eighteen Benediction (Shemoneh Esreh), which contains the phrase "You graciously bestow knowledge to man" (Cohen 1995, p. 90). Hence, creation and moral consciousness are two realms shared by the one God and humans. According to Cohen, this correlation is expressed in the songs of the Psalms: "sing neither of God alone nor of man alone" (Cohen 1995, p. 58).

### 7. Prayer as a Dialogue in Monologue

Cohen defines prayer as the soul's dialogue with God, which is "constituted by the monologue of the prayer" (Cohen 1995, p. 373). This wording, "dialogue in monologue", can be understood in several ways. It illustrates that the correlation with God, even though it is not perceived by the senses, is nevertheless possible. The prayer is a monologue because the central question is not whether God hears the prayer but rather the emphasis is on the person praying.[38] This also plays a role in his modification of Kantian ethics. Cohen writes that "The prayer secured the basic form of religion: the correlation of God and man. [...] the individual is now not only an element of totality, the symbol of mankind [as it is in Kantian Ethics], but his moral nature, as obtained in the prayer, is to himself, as it were, an absolute individual" (Cohen 1995, p. 376). Prayer establishes the self-respect of a person as a unique individual and not only as an agent of moral reason.

In addition, prayer as a "dialogic monologue" of the soul with God generates the individual's moral forces. Cohen writes

> "To begin with, this preliminary stage for the prayer, too, is of a purely moral character. For all spiritual, for all moral action, the mind needs to withdraw into itself; it needs the concentration of all its inner forces and prospects. As the solitude of the soul becomes a necessity in opposition to the whirl of sense impressions, so the soul psychologically is in need of withdrawal into itself, into its most inner depth, if it is to rise to the dialogue with the godhead. Prayer must be such a dialogue when it has to express in language confidence in God". (Cohen 1995, p. 372)

Similar to Kant's categorical imperative, which is formulated as a monologue (in which the person sees themselves as part of humanity), the prayer is also a monologue because God does not answer and does not take part in a conversation. However, unlike in Kantian ethics, Cohen conveys that the dialogic element is manifested in the idea of possible atonement from God. Only through reducing the ego and engaging in a process of "withdrawal into itself", which Cohen views as an act of modesty and humility in which "the I itself, the subject, becomes the object" (Cohen 1995, p. 373), is atonement achievable. On the one hand, although Cohen's style includes a certain form of pathos—a pathos according to which the I becomes the object and God becomes the subject—Cohen does not believe that God anthropomorphically forgives; rather, God is the notion that true and meaningful atonement is possible. In addition, he clearly states that the correlation with another human being must precede the correlation with God (Cohen 1995, pp. 114, 132). Nevertheless, and unlike in Kantian ethics, Cohen believes that the human being cannot resolve their consciousness of guilt without the inwardness (Innerlichkeit) of the prayer, an inwardness that connects them to their own subjectivity and hence to transcendence. In other words, the renewal of moral consciousness requires dialogue, which is found in prayer as "the psychological form of the religious factor of reconciliation" (Cohen 1995, p. 373). As such, prayer is a way in which a person examines themselves in front of something higher than themselves. The prayer expresses the individual's self-judgment before God; it expresses the individual's honest and deep confrontation with their consciousness of sin.

Without prayer as "the activity of language in which the will becomes active", argues Cohen, the thinking about *correlation* would remain theoretical (Cohen 1995, p. 399. See also his discussion about "God looks into the heart", p. 168). In contrast to the Hegelian dialectical synthesis, Cohen returns to the concrete religious language and revives it as a necessary factor in the foundation of moral consciousness. He highlights Psalm 86:11, "let my heart be undivided", in this context: "The unity of consciousness is the highest problem of systematic philosophy. [. . .] the prayer [. . .] becomes the linguistic means that continuously secures and establishes anew the unity of consciousness" (Cohen 1995, p. 379).

As mentioned, Cohen focuses mainly on the individual and not on public prayer. In his perspective, there is no room for prayer that does not originate in moral will, meaning that prayer should not be motivated by, or directed toward, one's own personal interests. At the same time, true prayer—that is, for the kingdom of heaven and the renewal of moral consciousness—cannot be imposed on the individual. Nevertheless, the inwardness of the prayer and confession have a communal context. Cohen claims that the inwardness of the prayer creates the individual, and only true individuals can form a plurality, as opposed to uniformity and totality (Cohen 1995, p. 376). Namely, the inner form of confession directs the outer form of the congregation, which is true plurality.

Prayer, as a form of dialogue with God, is accordingly associated, from the perspective of the history of ideas, with the evolution of moral consciousness in relation to the other person. In Cohen's terms, the development of the nature of the relationships with God (i.e., the development from materialistic sacrifice rituals to a direct unmediated correlation through prayer) leads to the development of relationships between human beings. He writes "God must become the creator a second time [. . .] he teaches man himself to create man as fellowman" (Cohen 1995, p. 147; Amir 2005).[39] These reciprocal relations are explicitly formulated "It is even a question, as yet not asked, whether I myself already do exist before the fellowmen is discovered" (Cohen 1995, p. 142). Namely, as the human being is created by God, the evolvementevolution of the moral self is, in a similar manner, conditioned on the formation of a moral relation with a particular "You".

## 8. The Historical Dialectical Development of Prayer

As noted, Cohen's neo-Kantian philosophy opposes the historicist and dogmatic approaches that developed after Hegel and that were also prevalent in Jewish studies of his time. Already in his *system of philosophy*, Cohen criticized Hegel's dialectical method as a means for comprehending history and acquiring ethical knowledge. In *Ethics of Pure Will* (published in 1907), he writes "The developmental perspective dominates Hegel's way of thinking. Dialectical movement is nothing other than development, and it is only too clear, everywhere in Hegel's thought, that the final result is always presupposed" (Cohen 2021, p. 38).

Hegel's approach was seen by Cohen as a "pantheistic error" in which the relationship between *what is* and *what ought to be* is lost. According to Cohen, Hegel's historical approach sees the present entirely in light of the past, and lacks the horizon of the ideal moral future and its influence on the establishment of the moral task in the present. In the pantheistic view (Cohen 2021, pp. 38–39, 93),[40] the future is "always presupposed". This expresses false metaphysics, which is also present in the mythological and idolatrous approaches (Pinkas 2020a; Bienenstock 2012). In *Religion of Reason*, Cohen claims that the genuine "concept of history is a creation of the prophetic idea [. . . that] the prophets are the idealists of history" (Cohen 1995, pp. 261–62). In other words, the prophets' struggle with mythological approaches embodies a pre-philosophical basis for the negation of a dialectical (passive and deprived of divine grace) approach to historical development. Moreover, processes of demythologization are not only a necessity for the advancement of the religion of reason but also characterize its development, and this process of demythologization gains momentum in Rabbinic Judaism.[41]

As for the question of the emergence of monotheism, Cohen's position is not completely determined. Sometimes, he refers to the emergence of monotheism as a wonder:

"From the historical point of view, which demands evolution everywhere, monotheism is and shall remain a mystery. No people and no spirit on earth had thought of the unique God" (Cohen 1995, p. 243). Namely, monotheism is not a product of historical development. Rather, it should be seen as a unique appearance—a revelation. However, in other places, he writes "The further development of polytheism leads to its self-dissolution (Selbstauflösung) in monotheism. [...] Monotheism [...] has its precondition in polytheism" (ibid, p. 376. See also there, pp. 36, 99, 340).[42] Cohen describes the historical development from polytheism to monotheism in terms of changes in the form of worship. Namely, human sacrifice was replaced by animal sacrifice (e.g., as reflected in the story of the binding of Isaac. Cohen 1995, pp. 171, 397) and the sacrificial rituals that characterize polytheism were replaced by monotheistic prayer rituals. This idea is also reflected in midrashic literature: "Israel said, "Master of the world, at the time that the Temple existed, we would offer a sacrifice and be cleansed. But now all we have in our hand is prayer" (Numbers Rabbah, 18:21). In Cohen's description, this evolution includes Hegelian dialectic characteristics, although this is not explicitly stated. He writes

> "If there were no prayer, worship would consist only in sacrifice. It is therefore possible to say that sacrifice could not have ceased if prayer had not originated in sacrifice and from sacrifice. [...] Prayer is an original form of monotheism. Of course, in this case too, as in that of all monotheistic creation, the general principle of any historical religious development holds true". (Cohen 1995, p. 371)

According to Cohen, prayer developed at the same time as sacrificial worship and gradually gained central status with the decline in sacrificial worship, especially after the destruction of the Temple. However, he assumes that due to the prophets' opposition to sacrifices, the transition from a temple to "a house of prayer" (Isaiah, 56:7) would have occurred even without the destruction (Cohen 1971, p. 222: "The prophets' zeal against these practices suggests to historians the possibility that the inner development of Judaism might have led to gradual abolition of those sacrificial rites even if the Temple had not been destroyed"). He uses the Day of Atonement (Yom Kippur) as a prime example of this inner development. This festival originally involved sacrifice rituals as a means of purification but eventually turned into a pure day of prayer and fasting. Cohen asserts that although atonement for ceremonial transgressions is mentioned in rabbinic literature, the focus is on moral offenses.

The dialectical element appears not only in the inner development according to which the act of sacrificial offering is replaced by a higher form of worship in prayer (the sensual worship transformed to a linguistic worship) but also in the fact that prayer preserves within itself the original notion of sacrifice: "The prayer is to replace sacrifice in order to achieve reconciliation. The broken hearts take the place of the slaughtered animal. Hence, humility originates in the correlation of man with God" (Cohen 1995, p. 394). That the explicit rejection of sacrificial worship by the prophets (i.e., self-negation) led to the rise of prayer expresses a dialectical development in which the advanced stage includes the previous one within it: "With regard to the sacrifice, the history of prophecy proceeds in two ways. One takes the road of the rejection of the sacrifice; the other, however, aims at its transformation; the alteration becomes transformation" (Cohen 1995, pp. 175–76).[43] Prayer as "an original form of monotheism" can be called "original" only if it preserves the essence (but not the form) of the original way of worship.[44] That is, in a certain sense, prayer already existed in the ritual of sacrifice in a symbolic, pre-conceptual way. At the same time, prayer underwent a stylistic and conceptual development, "from the praise of God into the longing of love for God" (Cohen 1995, p. 213).

According to Cohen, the idea of sacrifice—giving, giving up something, or submitting to something external to oneself, not out of convenience or self-interest but out of a sense of obligation—is a core concept that remains embedded in worship from sacrificial rituals to prayer and even in modern non-religious ethics. He writes:

"Among the wonders that are pertinent to the historical understanding of the wonder of monotheism, the fight of the prophets against the *sacrifice* occupies perhaps the first place. The entire classical world is attached to sacrifice; the idea of sacrifice is also the foundation of Christianity and, finally, one finds that this idea has also remained active in the most diverse modifications in the more free, modern consciousness. Not only every misfortune, but even every supposedly free moral action, is still understood as a sacrifice, if not to fate, then at least to duty. If one considers all this, it is almost incomprehensible how the prophets knew how to take superstition and paganism by the horns and how they recognized in sacrifice the root of idol worship". (Cohen 1995, p. 171)

When we consider the dialectical progression from sacrifice-based worship to prayer, we can assume that this movement will ultimately result in the sublation of religious symbolism with philosophical concepts (as Kant proposed in the context of the autonomy of moral duty vs. the heteronomy of prayer, and Hegel in the context of philosophy vs. religion in general). However, Cohen does not believe that the categorical imperative, which constitutes the ethical demand, is more effective than the language of prayer in achieving morality.[45] For Cohen, prayer is a dialogical monologue that preserves the idea of moral duty in a concrete way (i.e., as a practical moral reason) rather than in the abstract categorical imperative of ethical philosophy. Cohen's student, Franz Rosenzweig, even believed that the categorical imperative expresses an idolatrous regression rather than a monotheistic development because it expresses a withdrawal back to the worship of the human being themselves as a god.[46]

To conclude this part, I would like to underline the assertion that Cohen's view should be seen as a cornerstone for positions that (a) perceive prayer as the origin of dialogue, and (b) place dialogical reasoning as alternatives to dialectical approaches. Indeed, Cohen's historical analysis of prayer clearly expresses the application of Hegelian dialectical reasoning. In fact, according to Cohen, the development of Judaism into a religion of reason is based on processes of demythologization that are not free of dialectical reasoning. His student, Franz Rosenzweig, rightly crowns his teacher with the title of the "unconscious successor of Hegel", adding that precisely this unconscious Hegelianism was nonetheless justified (see Pinkas 2023, p. 106).[47] On the one hand, Cohen's historical account of the development of the religion of reason expresses the dialectical principle. On the other hand, the core of his neo-Kantian project is the rejection of historicism in general.[48] Namely, there is no historical process or dialectic behind human beings that makes them act out of necessity to bring forth the messianic age. Despite using Hegelian dialectic, Cohen in general takes a position that favors correlative (dialogical) relations over logical-dialectical ones, even if he does not explicitly state this as a *methodological change*. Franz Rosenzweig, who evidently established the dialogical approach in contrast to the dialectical method, is Cohen's successor in this regard.

## 9. Rosenzweig's Negation of *the Unity of Reason*

The existential-theological philosophy of Franz Rosenzweig (1886–1929) is an essential milestone in the development of a philosophical approach that focuses on relationships and dialogue, and positions itself as a *methodical* alternative to a logical-dialectical approach. Rosenzweig was well versed in the idealist tradition of German philosophy and became its most ardent opponent. He devoted his first book to Hegel's political philosophy (Hegel und der Staat, published in 1920). Following this, he felt an urgent need to overcome the idealist philosophical tradition "from Ionia to Jena" (Rosenzweig 2005, p. 18), from Parmenides to Hegel. Rosenzweig's existentialism is a counter-reaction to the philosophical tradition of German idealism and its methods that seek a knowing of *All* by means of logical reasoning alone. In Rosenzweig's thought, the *miracle of revelation* is a relational redemptive experience that in its essence refutes the absoluteness of dialectical reasoning. His existential methodology includes a transition from recognition (Erkenntnis) to experience (Erlebnis) in the quest for knowledge and truth.

Rosenzweig's explicit criticism of idealistic philosophy and his attempt to overcome idealistic absolutism are mainly discussed in the first part of his magnum opus, *The Star of Redemption* (published in 1921), but from a terminological and structural point of view, these ideas are present in the system of *The Star* as a whole. For example, in the second and third parts of *The Star*, he highlights a relational approach that includes prayer. Rosenzweig develops his ideas out of this confrontation, and although he offers an alternative, his thought involves the adoption of Hegelian insights. Namely, he refers dialectically to Hegelian philosophy while negating its terms (Amir 2004, p. 42; Mendes-Flohr 1992, p. 190).[49] Nevertheless, he offers a reversed refutation of Hegel because, for Rosenzweig, the philosophical realms (of logic, ethics, and aesthetics) negate themselves and are sublated into the concrete articulation of the theological realms (of creation, revelation, and redemption, respectively), which he presents as pathways for relationships between God, the human being, and the world (Rosenzweig 1999, pp. 70–71). Idealism (i.e., the "old philosophy"), claims Rosenzweig, with its unity of thinking (that everything can be known by reason alone) and its dialectical method of negation—which refers to God, the human being, and the world—is an erroneous attempt to reach an absolute truth about the essence of all things. One example of such an attempt is the negation of the *particular* exteriority for the knowledge of the internal *universal* essence or principle. Rosenzweig rejects this approach because it ignores the complexity of reality, and denies the existence of three elements—God, the human being, and the world—which are fundamentally distinct from one another, transcendental to each other, and can be known only from their own "revelation" in relation to the other element (Rosenzweig 1999, p. 75. See also his discussions in The Star concerning "Truth is not God. God is truth" (Rosenzweig 2005, pp. 403–18)).

The revelation of the three elements is possible as a form of a *relational event* and not of knowledge as a product of dialectical reasoning: "only in their relationships, only in creation, revelation, redemption, do they [God, the human being, and the world] open up" (Rosenzweig 1999, p. 85). The "All", writes Rosenzweig, must be grasped "beyond cognition [i.e., the dishonest cognition of idealism] and experience [i.e., the obscure experience of the mystic] [...] this grasping takes place", as Rosenzweig writes in the third part of *The Star*, in "the illumination of prayer" (Rosenzweig 2005, p. 414). The choice of the term *illumination* of prayer is not accidental; it contrasts with the term "light of reason", which has characterized philosophy since the early Enlightenment. Instead of the *unity of reason* (the dialectical synthesis) expressed in philosophy, Rosenzweig offers the *unity of God*. Instead of the light of reason, Rosenzweig offers the illumination of prayer. Instead of *knowing* the absolute, he offers *speaking* and relations (Neeman 2016, p. 194). For him, the unity of thought is abstract, alienated, and deceptive, while true prayer can lead to actual unity.[50]

Rosenzweig does not mean to imply that prayer allows one to instantly perceive all of existence by negating cognition and experience. Rather, he claims that the illumination of the "All" must originate from a point of eternity that is beyond cognition and experience, though it may still be reflected through them (Turner 2014, p. 189). Generally, just as prayer, according to Hermann Cohen, is a mediation in the *correlation* between the human being and God, for Rosenzweig, true prayer correlates between the miracle of creation and revelation, and also anticipates future redemption.

Rosenzweig's thought is not a medieval-style polemic in which philosophical analysis promotes theology. Instead, philosophy is necessary to elucidate the underlying presuppositions of theology (Schwarz 1978, p. 242). This leads to the combination of philosophical thought and theological belief into "new thinking" (a term that refers to both philosophy and theology): "The theological problems are to be translated onto the human, and the human driven forward until they reach the theological" (Rosenzweig 1999, p. 89).[51] It can, indeed, be argued that this combination is achieved through a dialectical synthesis (sublation of the old into the new thinking). However, the *relational* conclusions of his method oppose this constellation.

### 10. Prayer and Language

Rosenzweig's starting point is the post-Enlightenment atmosphere, where on the one hand, prayer in its traditional form is not relevant to the ordinary person, not even in light of its rational and moral elements; on the other hand, in light of "death and love", prayer is revealed as an existential necessity. He defined *The Star* as "a system of philosophy" and not a philosophy of religion (Rosenzweig 1999, p. 220).[52] That is, theological terms and events play a role in a broad philosophical outlook. Accordingly, his discussion of prayer goes beyond conventional religious discourse. Prayer for him conveys both a religious and an existential (theo-psychological) need. On the one hand, everyone prays, even if they are not aware of their prayer nor of its motivation, style, timing, and intended recipient. On the other hand, prayer is a component of the common religious lifestyle and is included in the liturgy in prescribed formulas and times. Ehud Neeman argues that the transformation that Rosenzweig went through from a relativist intellectual to a person who recognizes the importance of faith and the reality of God as a living and revealing force occurred due to his recognition of the power of prayer (Neeman 2016, p. 141).[53] Neeman claims that the wisdom of prayer, or "pray-sophy", according to Rosenzweig, is an alternative to philosophy and that human completion is only achieved through relations with God (Neeman 2016, pp. 194–95). Indeed, Neeman considered prayer to be the heart and purpose of *The Star* (Turner 2014; Amir 2004, p. 197).

The subject of prayer demonstrates the complexity of Rosenzweig's relationship to the Hegelian dialectic. On the one hand, his philosophy and especially the method of *the* "new thinking" demonstrate a methodical criticism of the Hegelian dialectic and its desire for an *idée fixe* (Rosenzweig 1999, p. 83). On the other hand, he uses dialectics to analyze the content of prayer itself. Consequently, Rosenzweig's attitude toward prayer should be examined within the context of his philosophy of language.

For Rosenzweig, language is a fundamental category of reality. That is, he assumes an identity between language and actuality, a position that stands in contrast to the "identity of being and thinking" that characterizes idealism (Rosenzweig 2005, pp. 18–19, 24, 61). According to Rosenzweig, the new thinking brings a new method: "The [new] method of speech takes the place of the [old] method of thinking" (Rosenzweig 1999, p. 86). The grammatical "speech-thinking" (Sprachdenken) replaces the logical "think-thinking" (Denkdenken). The term speech-thinking implies opposition to a think-thinking approach, which starts with an abstract thought seemingly unrelated to anything (an independent cogito). Speech-thinking, however, is rooted in reality; its starting point is words, the speakers of these words, and tangible situations. He writes "The new philosophy does nothing other than turn the 'method' of common sense into the method of scientific thinking" (Rosenzweig 1999, p. 83, and see also especially on p. 87). The difference between the two methods lies only in "the need of another" (Rosenzweig 1999, p. 87; See also Horwitz 1981, p. 26; Pöggeler 1984). The "I" needs others for his realization. The old idealistic philosophy ("think-thinking") analyzes the human being as a singular, while speech-thinking views human beings in terms of their relationships and associations with others. The connection between prayer as a dialogical model of witnessing God's love, which leads to intimate recognition of the other, and the project of redemption appears explicitly in Rosenzweig's thought. He writes "for what else is redemption but that the I learns to say you to the he?" (Rosenzweig 2005, p. 292).

According to Rosenzweig, the methodological change from think-thinking to speech-thinking refers to human language as a whole, not just to theological discourse. However, this becomes clearer in light of the differences between theology and philosophy, in their use of language and their positions regarding it. The "old philosophy" speaks in terms of logic and mathematics and expresses doubt about the truthfulness of the spoken language. He writes "Idealism tries to elevate itself above language with its own logic that is hostile to language" (Rosenzweig 2005, p. 152), and "Idealism rejected language as organon [a means of establishing knowledge]. [...] Idealism lacks naïve trust in language. [...] It asked for reasons, justifications, and forecasts, everything that language could not offer it,

and for its part it invented logic, which provided all this" (Rosenzweig 2005, p. 157). In contrast, in theology, language is "the organon of revelation" (Rosenzweig 2005, p. 120); speech is the "tool" with which God created the world, and speech precedes creation because "God spoke" (Rosenzweig 2005, p. 123).[54] In theology, according to Rosenzweig, the origin of language is God. Language and speech precede the human being; this is what "makes of man a man. [...] language is truly the wedding gift of the Creator to humanity" (Rosenzweig 2005, p. 120; See Horwitz 1981, p. 27; Schwarz 1978, pp. 269, 278). Rosenzweig aspires to found a theological existentialism that combines the knowledge of philosophy with "trust in language", as it is in theology. In his approach, philosophy and theology are "siblings" rather than two distinct disciplines. On each side, there is a deficiency that only the other side can fill (Rosenzweig 1999, p. 89).

Language plays an important role in all three parts of *The Star*. In the first part (creation), the language is the language of philosophy, logic, and mathematics, which deals with describing the past (e.g., the language of creation is a descriptive language in which the Creator is hidden, distant, and impersonal). In the second part of *The Star* (revelation), the language is the language of love, which occurs only in the present and is perceived as an imperative. The language of love appears in an intimate way from "a lover to his beloved" (Rosenzweig 2005, p. 175). He writes, "With the call of the proper name, the world of Revelation enters into real dialogue", Rosenzweig 2005, p. 201). He writes, "The commandment of love can only come from the mouth of the lover. Only the one who loves [...] can say and does say love me" (Rosenzweig 2005, p. 190; See Batnitzky 1999). In Rosenzweig's view, it is only because God loves that humans are able to love (Bergman 1955). God's love is the heart of the dialogic event between God and the human being, and this enables praying. The ability to pray is "the greatest gift given in revelation" (Rosenzweig 2005, p. 198), and this ability, according to Rosenzweig, becomes an existential necessity. Prayer is referred to as a dialogue with God that take places in the present moment. Hence, prayer emerges spontaneously as a confessional response to the revelation of God's love (see Pinkas 2020b), which Rosenzweig describes in this context as "a call to hear", as expressed in the most important prayer in Judaism, the Schema Israel (Deuteronomy, 6:4: "Hear, O Israel: the Lord is Our God, the Lord is One"). By hearing the "voice of love" of God, the solitary self opens up and becomes "a soul that speaks" (Rosenzweig 2005, pp. 88, 182). Rosenzweig defines this transformation as "from the miracle to the illumination" (Rosenzweig 2005, p. 280). Revelation is the emergence of divine speech in the human soul, and prayer is the soul's reaction to this speech. It is an authentic expression of the soul that becomes aware of the revelation of God's word. As the person experiences the revelation, the prayer becomes a certainty—a desire for a dialogical partnership with God.

In the third part of *The Star*, the language of the liturgy (e.g., public prayer and gestures of worship) is methodologically presented as a mediation between revelation and redemption. Compared to the informative a priori "silence" of the "mathematical symbols" with regard to the past *miracle* of creation, and the "grammatical forms" of the present dialogue in the miracle of revelation, the "liturgical forms" are an anticipation of the future redemption. The liturgical forms "anticipate; it is a future that they make into a today" (Rosenzweig 2005, p. 312). Anticipation is the power of prayer, and unlike other languages, this can bring eternity into the present moment (Amir 2004, p. 144).

According to Rosenzweig, there are two central characteristics of prayer: First, individual prayer is the spontaneous prayer that characterizes the course of revelation—that is, the human being's response to God's love, which is felt as an imperative to return love to the lover, as demonstrated in "You shall love the Lord your God" (Deuteronomy 6:5). Second, public prayer is a continuation of individual prayer but conducted in the community; it aims to fulfill the commandment to love God that was given to the individual by addressing the world, "Love your fellow as yourself" (Leviticus 19:18). Hence, prayer, according to Rosenzweig, is both the deepest personal expression of the individual and the noblest expression of society. He emphasized the daily importance of prayer and saw the

prayer book (the siddur) as the essence of Jewish life.[55] Neeman suggests that Rosenzweig found in the prayer book a replacement for Hegel's *Introduction to the History of Philosophy*. Hegel showed that truth was formed from the accumulation of ideas over the generations. Similarly, Rosenzweig regarded the prayer book as a book that treasures the essence of revelatory truth in Jewish life (Neeman 2016, pp. 194–95).

In *The Star*, the central discussion of prayer is in the third part, which discusses redemption; it opens with the title, "On the possibility of obtaining the Kingdom by prayer" (Rosenzweig 2005, p. 283). Rosenzweig deals with prayer as the central action of the person who receives revelation and is called to redeem the world. "Prayer establishes the human world order" (Rosenzweig 2005, p. 286) and places the human being in light of eternity and as part of a community. According to Rosenzweig, each religion's representation of the "kingdom of God" is embodied in prayer, which he views as a vital tool for defining space and time. True prayer enables God to intervene in human history without being constrained by the (dialectical) laws of history. According to Rosenzweig, this is embodied in the liturgy, which on the one hand guards against the identification of God with historical processes, and on the other guards against the denial of God's involvement in the world. In the last part of *The Star*, Rosenzweig maintains that the human being finds themselves within the truth—as a part of a community and its liturgical gestures (the collective liturgical formulations of longing for the truth)—a unity that is beyond words. In his view, without liturgy in general and prayer in particular, the human world would remain intellectually and spiritually incomplete.

## 11. The [Dialectic] Ability to Pray

Rosenzweig, like Cohen, saw prayer as the heart of religious life and sought to understand its meaning for the common person in modern times. Both Cohen and Rosenzweig did not deny the halachic status of prayer, although it does not serve as a theoretical starting point. Rosenzweig saw prayer as related to life in general,[56] and, unlike Cohen, did not reduce his interpretation of the religious experience to its ethical quality. The prayer, says Rosenzweig, is "reached in revelation" (Rosenzweig 2005, p. 198); as such, its purpose is for its own sake. The purpose of prayer is to develop the ability to pray: "For with the gift of the ability to pray (Betenkönen) an obligation to pray (Betenmüssen) is imposed upon the soul" (Rosenzweig 2005, p. 199). The obligation to pray can be interpreted as an uncompromising and basic need for truth and illumination that grasps the *All*.

This raises the following question: If authentic prayer is possible based on the recognition of *the miracle of revelation*, how can it be applicable to ordinary people for whom revelation is not feasible? Rosenzweig's starting point is the modern person who does not see prayer as relevant or possible. In his method, the concept of *miracles* is an epistemological category that is distinct from both philosophy and theology and, as such, serves as a link between these two realms. Acknowledging the *miracle of creation* is an epistemological prerequisite for experiencing the *miracle of revelation*. However, this acknowledgment should also be anchored in the experience (Erlebnis). He therefore links the possibility of experiencing the miracle of creation with death. "Death" is the "keystone of creation" (Rosenzweig 2005, p. 169). The fear of death is the primary factor that puts the human being in front of themselves and shakes them from their slumber.[57] Rosenzweig quotes, "For love is fierce as death" (Song of the Songs 8:6), to emphasize that death is the point of origin that binds the human being to creation. The fear of death leads to the recognition of creation as a recurring act of God's love. In relation to the world, God's love results in creation, while in relation to humanity, God's love is the possibility of revelation and, hence, prayer. In this sense, the ordinary non-religious individual (a closed Self that does not yet recognize itself as a soul that speaks) has a better dialectical starting point to arrive at authentic prayer than the religious person. For the non-religious individual, prayer does not come from their sociocultural environment but emerges from their own existential experience of fear and love. Bergman explains that revelation paradoxically awakens the divine aspect of human existence in so far as it sets aside the entropy of forces that

ultimately distance humanity from God (Bergman 1955, in Turner 2014, pp. 176–77), as expressed in the declaration, "All Revelation begins with a great No" (Rosenzweig 2005, p. 187). Hence, the dialectical element of prayer is the negation of its impossibility. The authentic starting point of the modern, ordinary human being is that God is hidden. But the certainty about death and love leads to an inner transformation that makes prayer possible and henceforth necessary. Prayer is an *existential* necessity (not in the religious-institutional sense) because it means recognition and response to His or Her *love* for me. This response transforms the Self (which fears death) into a beloved soul (which believes in miracles). Consequently, authentic prayer is a task that cannot be learned but is merited; it is the expression of human longing, on the one hand, and God's love, on the other.

Rosenzweig begins the third part of *The Star*, titled "On the possibility of obtaining the kingdom by Prayer", with the statement, "That one might be able to tempt God is perhaps the most absurd of the many absurd assertions that faith has brought into the world" (Rosenzweig 2005, pp. 284–85). Following this, he presents a dialectical analysis of a verse from the Jewish (morning blessings) and Christian (Matthew 6:13) petitions, "Lead us not into temptation!"[58] Traditionally, this petition is an acknowledgment of the human being's weakness and hence is a request to God to not test one's faith, morality, etc. Rosenzweig seemingly reverses the equation and sees this petition as tempting God. Indeed, the basic premise of the petition expresses a request to God, and not gratitude or praise. Rosenzweig claims that this petition is problematic because it expresses the absurd idea that God, who is beyond all things, can be subject to human wishes. Namely, how can God be subject to something as limiting and human as temptation? Rosenzweig claims that this petition is a "twofold denial of his providence and of his fatherly love", a denial of God's limitless power as a creator and ultimate freedom as the loving revealer. Rosenzweig's answer is that while God as a creator and revealer cannot be tempted, he can be tempted as a redeemer.

In redemption, the active agent acting on the world is not God, rather the human. To start with, the ability to pray is given by God (i.e., providence enables dialogue with God); in contrast, it brings with it the possibility to influence God—that is, to force him to reveal himself as a redeemer. Affecting God in the context of creation and revelation is "absurd" (i.e., no person can force God to create or to love). However, influencing God is possible on the path of redemption. On the one hand, redemption is the path of the relationship between the human being and the world: "For what else is redemption but that the I learns to say you to the he?" (Rosenzweig 2005, p. 292). On the other hand, when the human being fulfills a loving relationship with the world, then redemption is "self-redemption for him [God]" (Rosenzweig 2005, p. 290).

The foundation for Rosenzweig's dialectical analysis of this petition is found in the traditional Jewish commentaries on the *Book of Job*, which acknowledge that God can tempt humanity only if humans can tempt God (Rosenzweig 2005, p. 284). He writes, "This freedom of prayer shows itself in the possibility of tempting God, wouldn't then maybe the temptation of man by God be the necessary prerequisite of this, his freedom?" (Rosenzweig 2005, pp. 284–85). Rosenzweig claims that God's freedom depends on the human being who tempts him; in fact, only in this way can God be revealed.

The petition's premise also conveys the notion that an individual is prepared to renounce their autonomy and rigidly submit to God's rule. This is absurd, though, as the request itself admits that a human being has autonomy, at the very least to decide whether or not to fear God (BT, Berakhot 33b:23: "Everything is in the hands of Heaven, except for fear of Heaven"). That is, God needs humans in order to freely choose to obey his law, which is consistent with his will.

Rosenzweig brings an example from a rabbinic legend about the Sabbath laws that supports the idea that it is God's will that the Jews observe his Sabbath regulations, but they cannot be coerced into doing so.[59] That decision must be free, and it can only be free if the people making it do not think that he will reward them for obeying. Otherwise, it would not be a free decision; rather, it would be one that was motivated by some other

reasoned factor. Rosenzweig's dialectical reasoning of the prayer proceeds as follows: God tempts humans into believing there is no reward for obedience and that it could even result in punishment in order to give them freedom of choice. By wagering that following God would be more profitable than defying him, humans tempt God and disrupt his attempts to make their choice of obedience free, which is a necessary condition for the act to be one that redeems the world. Rosenzweig's dialectical reasoning of this petition is used to explain God's freedom as a necessary condition for human freedom.

In Rosenzweig's view, the dialectical idea of praying for the ability to pray and praying for the coming of the Kingdom (which is a prayer for the "future repetition" of the miracle of revelation, Rosenzweig 2005, p. 199) are the only two types of genuine prayer. Every other type of prayer, including prayers for rain, blessings for the year, the recovery of the sick, and the atonement for sin, is conditional and must be followed with the phrase, "May Your will be done and not mine" (Bergman 1955, p. 107; See Mishnah, Pirkei Avot 2:4 and NT, Luke 22:42. Rosenzweig sees this insight as common to Judaism and Christianity). This idea is reflected in his distinction between the magician, who wants to influence God with words of magic, and the true worshiper, who only wants to entrust his will to God's will (Rosenzweig 2005, p. 105).[60] It should be noted that an explicit formulation of the distinction between magic and monotheism based on the *dialogic* factor was expressed shortly after by Martin Buber.[61]

However, in Rosenzweig's thought, there is no such thing as "improper content" in prayer; every prayer begins with laying down the afflictions of the soul before God, and as such is fundamentally authentic and legitimate. Hence, prayer is appropriate or inappropriate with respect to "time" (Turner 2014, p. 185). Rosenzweig distinguishes between different types of prayer: the prayer of the believer, the sinner, the fanatic, and the unbeliever. In his view, these are qualities of prayer that can exist in the same person at different times. True prayer is the believer's prayer, which anchors the individual prayer within the public-communal prayer, and comes "always in good time [. . .] neither early nor late" (Rosenzweig 2005, p. 307). The sinner's prayer is a selfish prayer for one's own benefit; the problem with this "egoistical prayer" is not in the content of the request but rather that it "denies redemption" and strives to hasten the occurrence of earthly events. Ultimately, it "delays the advent of the kingdom" (Rosenzweig 2005, p. 292; See Turner 2014, p. 185). The fanatic's prayer is the opposite of the sinner's prayer because it imposes the future on the present and it attempts to control the world. The fanatic wants "to hasten the future of the Kingdom so that it might come ahead of time" (Rosenzweig 2005, p. 293). The prayer of the unbeliever expresses acceptance of destiny out of expectation and approval of the connection that exists between the human being and the world, but without praying for the kingdom. Each type of prayer has an epistemological orientation that is limited according to the ability to grasp the All. That is, the three elements (God, humanity, and the world) and their connections (creation, revelation, and redemption) are included in the "illumination" of true prayer (Rosenzweig 2005, p. 288). The significant difference between the prayer of the believer (e.g., Moses "the man of God") and the prayer of the unbeliever (e.g., Goethe "the man of life", Rosenzweig 2005, pp. 293, 314) is that the first takes place in public, thus completing the unbeliever's prayer (which is always only an individual prayer) and expanding it from the course of revelation (the dialogue of love between the individual and God) to the course of redemption (directing the dialogue of love from God to the world). To sum up, Rosenzweig claims that prayer contains the dialectic that holds that God is both inside and outside of the concrete phenomenon and the physical encounter. God is both outside of the world of phenomena as its creator, and inside of it, through the human soul, as a lover. God depends on humans since they are his means of redemption; as such, he permits temptation from them.

## 12. Philosophical Symbols and Dialogical Liturgy

Rosenzweig's reasoning, on the one hand, confirms the freedom of thought achieved in dialectical thinking. However, he refutes the assumption that everything in the world

can be perceived by thinking alone. In Hegel's philosophy, the dichotomies are overcome and all oppositions are logically unified. Hegel's totality of the All placed philosophical reasoning (the absolute spirit) above all other realms. In Rosenzweig's words, philosophy for Hegel is the "fulfillment of that which is promised in Revelation" (Rosenzweig 2005, p. 13)—that is, philosophy is a sublation of theology. Hence, the former substitutes the latter. Specifically, Hegel's method includes the realm of faith within the realm of reason. In contrast, for Rosenzweig, spoken language has priority over thought in representing the complexity of reality, and the illumination of prayer has priority over abstract thinking in grasping the All. This idea is sharpened in his introduction to the last part of *The Star*, where he asserts that the height of liturgy "is not in the common word but the common gesture", and that the liturgical gesture transcends language and becomes "something more than language" (Rosenzweig 2005, pp. 313–34).

Despite the importance of the spoken language in *The Star*, he compares the "mysterious silence" of the mathematical symbols with the silence achieved in the liturgical gesture. Namely, the silence of the philosophical symbols, which is "the 'a priori' heirloom of a pre-creation", is firstly revealed with the spoken language: "the world is never without the word, it would itself also not exist" (Rosenzweig 2005, p. 312). That is, the mystery of creation is uncovered through the spoken word of God in revelation. Secondly, the grammatical forms (in the course of revelation) are replaced by the silence of the illumination of the liturgical forms (over the course of redemption). He highlights, "Light does not talk, but shines" (Rosenzweig 2005, p. 313). Unlike the mysterious silence of the philosophical symbol, the silence of the liturgical gesture is one of understanding that no longer needs words and speech. The dialogue with God in the common prayer is a gesture beyond words.

As already mentioned, in Rosenzweig's method, philosophy and theology complement each other; the believer's prayer and the unbeliever's prayer, and the theologian's prayer and the philosopher's prayer are both necessary: "Divine truth hides from the one who reaches for it with one hand only [...]. To the one who calls to it with the double prayer of the believer and of the unbeliever, it will not be denied. God gives of his wisdom to the one as to the other, to belief as to unbelief, but to both only when their prayer comes jointly before him" (Rosenzweig 2005, pp. 314–15). That is, a liturgical gesture rather than dialectical reasoning unifies the prayers of the philosopher and theologian.

## 13. Concluding Remarks

This paper is founded on two philosophical assumptions: the first is that there is a difference between two patterns of recognition: the dialectical and the dialogical. The second assumption is that the origins of the dialogical pattern may be found in the relationship between human beings and God, a relationship in which *prayer* has a major role. The second assumption leads to the supposition that the emphasis of the dialogic approach on moral responsibility is theologically grounded. In other words, the relationship between humanity and God serves as a paradigm for human relationships.

The dialectical pattern asserts that dissimilarity causes tension (discomfort, aporia), and that in every confrontation, there is a more developed "winning side", which, in sublation, contains the preceding side within itself. The dialectical synthesis resolves the tension between the parties through an overall abstract unification. Hence, progress and fulfillment are constant movements toward a higher level of unity in thought. The dialogical pattern, however, asserts that both sides continue to contend and remain fundamental aspects of the same system without merging (see Meir's studies, for example: E. Meir 2013). According to the dialogic approach, the development and self-fulfillment of the subject depend on mutual relations with another being.

If we perceive the dialogic pattern as grounded in the relationships between the human being and God, then we will assume that a genuine dialogue preserves within it the weight of the original meaning of religious worship. Namely, the dialogue involves the element of offering sacrifice and prayer. Consequently, the dialogue, as a ritual, is "listening" to an

absolute otherness and "responding" to its call. In order to accommodate this otherness, the subject must lower their position with humility and modesty, which are central factors in prayer. The humility in religious rituals is preserved in human dialogue as a moral obligation and responsibility—the ability to respond to another being that essentially is beyond any of one's conceptualizations and projections. In addition, genuine dialogue brings change and transformation to the subject. Emmanuel Levinas formulated the idea that "the essence of discourse is prayer" in *Entre Nous, On-Thinking-of-the-Other*.

Focusing on Hermann Cohen and Franz Rosenzweig in the context of prayer and dialectic highlights the complexity of these themes in modern Jewish thought. These two philosophers utilize dialectical reasoning while also criticizing it and offering an alternative. Cohen returns to Kantian thought in order to correct German idealism and Hegelian approaches to understanding history. He is trying to restore trust in an a priori *good* that is not a product of dialectic but of *created* moral reason. For him, true prayer is a correlative event (between the individual and God, and between humans) that actualizes the social moral task. In its essence, prayer is a break from a dialectical historical view, which Cohen considered deterministic. Prayer is a moral obligation and a way to take action in the world, and the renewal of moral powers in the human being. However, Cohen's description of the historical development of prayer rests on dialectical reasoning: prayer developed dialectically out of sacrificial worship and replaced it. This demonstrates that Cohen did not completely renounce dialectical historical logic and employed it for his philosophical and hermeneutical purposes. Cohen's approach inspired the second assumption of this paper (mentioned above). Rosenzweig develops his existential thought as a negation of idealist philosophy. His *new thinking* proposes a methodological change from dialectic to dialogue, and his *illumination of prayer* offers unity, which is a substitute for dialectical synthesis. However, Rosenzweig's thought abounds in dialectical reasoning. This is clearly expressed in his opinion about the possibility of praying in general, and in his analysis of the meaning and content of the "temptation" prayer.

The difference between Cohen and Rosenzweig can be understood in light of their views on Judaism in general. In Cohen's approach, there is a focus on the harmonization of Judaism with the tenets of the religion of reason, universalism, and autonomy, which is achieved through demythologization processes that entail dialectical historical development. For Rosenzweig, the starting point is the existential experience: fear of death and the power of love. Life is the starting point from which the individual "returns" to tradition and to the scriptures, which are fixed and unchanging, and thus include eternity within them. For Rosenzweig, Judaism is metahistorical. The holy land, the holy language, the holy law, and prayer "can probably be evaded but not changed" (Rosenzweig 2005, p. 323). In the context of prayer and revelation, this seems to also be expressed in their different choice of ideal Jewish medieval figures—namely, Cohen turns to Maimonides' "ethics" and rationalism while Rosenzweig turns to Judah Halevi's faith and poetics (and liturgical Piut in general).

The conclusions of both Cohen's and Rosenzweig's (as well as Buber's and Levinas') thought, in general, and their position on prayer, in particular, demonstrate a preference for a relational way of thinking over a dialectical one, but without renouncing the latter. Although each express it slightly differently, these thinkers believe that spoken language has a theological foundation. The language of philosophy (logic, dialectics) is reflective, descriptive, and monological in nature. It brings order and clarity, but it does not demand the moral imperative and responsibility of listening and responding. This paper supports the position that the dialogical ties between theology and philosophy should be tightened.

**Funding:** This research was funded by the Deutsche Forschungsgemeinschaft (DFG, German Research Foundation), grant number Projektnummer 491466077.

**Institutional Review Board Statement:** Not applicable.

**Informed Consent Statement:** Not applicable.

**Data Availability Statement:** Not applicable.

**Acknowledgments:** This article is dedicated to Johannes Hildebrandt.

**Conflicts of Interest:** The author declares no conflict of interest.

## Notes

1    Plato coined the term "dialectic" to describe the discipline of philosophy, as opposed to sophism, which he did not perceive as dialectic. Nikulin argues that dialectic was originally an oral practice originating in oral dialogue; written dialogue then developed as an imitation of oral dialectic; and finally, written dialectic was refined into a non-dialogical and universal method of reasoning (Nikulin 2010, p. 2). Nikulin asserts that in modern philosophy, dialogue is ousted by the advent of the Cartesian, self-centered, autonomous, and universal subject; instead, a dialectic of philosophical analysis develops as the correct method for reasoning.

2    Spinoza is one of the early forerunners of modern thought and one of the first, already in the seventeenth century, to criticize religious belief. Regarding prayer, he writes ironically: "If prayers could help, then one ought to pray for his [the Devil's] conversion" (Spinoza 2002, p. 89). His statement, "He who loves God cannot endeavor that God should love him in return" (Spinoza 2002, p. 372), is one of the most influential sayings in the Jewish philosophy that developed after him.

3    Following Kant and, notably, Sigmund Freud, the premise that prayer is "wishful thinking" was prevalent in psychology and anthropology. In our time, Dawkins defines prayer as an irrational activity that characterizes the theist. More than showing that prayer has no real power, Dawkins seeks to show the illogicality of theological thinking about God. He writes: "To adapt Alice's comment on her sister's book before she fell into Wonderland, what is the use of a God who does no miracles and answers no prayers?" (Dawkins 2006, pp. 60–66). Modern criticism of prayer is based on a more general criticism that religions lead to separation and sectarianism. See also there, pp. 277–78, his discussion about the use of prayer in Nazi Germany, with the goal of building Nazism into a religion. In a similar way, Sam Harris presents the irrationality, absurdity, and moral risk that exist in the belief in the efficacy of prayer (Harris 2005, pp. 44–49).

4    According to Heschel, "religious behaviorism" is in large measure responsible for the crisis of prayer. See also Heschel (1966, pp. 320–21).

5    The most well-known example is probably Adorno and Horkheimer's criticism in *the Dialectic of Enlightenment* (published in 1947), which pointed out the moral problem in the historical dialectical approach. Hannah Arendt considered the dialectical approach responsible for the development of fantastic historical-political approaches: fascism and Nazism (Arendt 1976, pp. 468–72). Karl Popper's criticism of the dialectical method is also well known (Popper 1940). Popper concluded: "It would be best, perhaps, not to use it [dialectic] at all; we can always explain such developments in the clearer terminology of a trial and error development. [. . .] The main danger of such a mix-up of dialectic and logic is, again, that it offers help for arguing dogmatically".

6    E.g., (Wilber 2000). In my opinion, Wilber may be seen as Hegel's contemporary successor because, in his method, philosophy (that is, the language that organizes the link between art, ethics, and science) can replace religion.

7    Chamiel claims that dialectic characterizes modern Jewish thought and that many thinkers of the 20th century and the beginning of the 21st century found it appropriate to describe Judaism as a *dialectical religion* that essentially offers a synthesis of the dialectical tension between reason and faith. Chamiel coined the term "dialectical believer" and attributes it to the rational believer who proposes a reconciliation between the monotheistic belief in the world of God and the (scientific) perspective of separated domains. In my opinion, Chamiel is right in his observation that contradiction, perplexity, and doubt are the vital driving forces behind the rational person's creativity, and that "they allow mankind to evolve and progress" (Chamiel 2020, p. 202). However, in my view, and as we can see in Buber's thought and thereafter, the premise that a contradiction between two different notions stands necessarily in a "dialectical relationship" is an incorrect assumption. Not every contradiction and opposing set of opinions expresses dialectic. In Hegel's philosophy, the dialectic movement is preliminary a moment within the subject itself, and not between two different subjects.

8    See his fascinating words there about "the polarity of Judaism". See especially Heschel (2005, pp. 708–10): "Torah can only be acquired in two ways: with reason's lens and the heart's lens. One who is blind in one eye is exempt from the pilgrimage. [BT, Hagigah 2a]. [. . .] Negative statements have positive connotations, and vice versa. Thought develops only through dialectic— through the synthesis of concepts that are opposed to one another and complement one another. A knife can only be sharpened by the blade of its counterpart. And here is a precious principle that was articulated by our Rabbis: "A controversy that is for a heavenly purpose will in the end endure" [Mishnah Avot 5:17]. Thus, whoever says that these two approaches contradict one another is simply mistaken. Both are focused on one reality, and each is subsumed by the other. The hidden essence of reality is that of two natures coming together. [. . .] Despite the appearance of contradiction, there is in fact a covenant between opposites, a covenant that unites different modes of apprehension". See also (Luz 1982).

9    This is a far-reaching claim that deserves in-depth study. Can a synthesis of opposites (God is near and far, etc.) fulfill the mental-spiritual role that religious dogma plays? Is a *dialectical religion* concretely possible? Thinking about Erich Fromm's term for Religion, one can ask: what is the *frame of orientation* that such a religion can offer? What is the concrete *object of devotion* that it can present? Hence, what does faith mean when the object of devotion is a contradiction in itself?

10   Heschel considered Buber's thought a form of philosophical anthropology. See Even-Chen (2005).

11   See (Chamiel 2020, pp. 68–78). See also (Kaplan 1996, p. 62, fn. 4), about the weaknesses of Heschel's dialectic, which Kaplan describes as "unconvincing by normal philosophical standards".

12   Since the beginning of the 20th century, anthropological studies have emphasized the universality of prayer as an action that transcends certain times and cultures. Psychologists such as William James believed that prayer is a natural human tendency, and does not characterize only the religious consciousness (e.g., James 2002, pp. 357–69).

13   Rosenzweig maintains that Hegel's inquiries into the origins of knowledge and authority are dialectical and hence speculative and cannot, therefore, result in freedom.

14   I introduce Buber's dialogical philosophy without investigating the specific implications of Cohen and Rosenzweig or other notable influences, such as Hasidism, Kierkegaard, Feuerbach, and Eastern mysticism, on his dialogic thought, which is certainly important but stretches beyond the scope of this study.

15   Buber distinguishes between "I-It" and "I-Thou" relationships. In I–It relations, the appeal to the other is "natural" (as it is usually in social, scientific, educational, clinical, economic, and political relationships), in which the status between the two parties is not symmetrical (one side is seeing the other in light of a goal that is beyond the dialogue itself). I-It relationships are used for benefit and service relations. The I–Thou relationship is a unique dialogue in which there is a renunciation of the layers of the external identity and a meeting of the other from within the depths of the being. An I–Thou encounter is unique, and as such, it is the revelation itself. For Buber, an I–Thou relation is not only the foundation of moral reason (as Cohen thought), but the existential status of all beings in general. Buber's dialogical philosophy is anchored in theological thinking (e.g., the idea that the soul exists in the upper worlds and maintains a dialogue there with God before the birth in the lower world takes place; Midrash Tanchuma, Pekudei 3:6; Zohar 3:13a:9; Nachmanides on Genesis 2:7:1).

16   See also Hegel's words about the "speculative" (which contains the opposing sides) as synonymous with the mystical: "But as we have seen, the abstract thinking of the understanding is so far from being something firm and ultimate that, to the contrary, it turns out to be constantly sublating itself and changing over into its opposite, whereas the rational as such consists precisely in containing the opposites as ideal moments within itself. Thus, everything rational is to be called at the same time 'mystical'". (Hegel 2010, p. 133).

17   According to Cohen, the Logos allegedly replaces the belief in one God, thus damaging the possibility of a direct correlation between the human being and God. He writes: "The Logos [. . .] become a second God, and yet there is no first, but only the one unique God. [. . .] The Logos [. . .] suffers from a basically erroneous idea. It overrated the importance of *existence* with regard to nature and the human spirit" (Cohen 1995, p. 48. See also on pp. 100, 201, 239).

18   Admittedly, dialectical logic can be found in Buber's *I and Thou*, for example, by placing I–It as opposite to I–You relations. He writes: "The World is twofold for man in accordance with his twofold attitude. [. . .] One basic word is the word pair I-You. The Other basic word is the word pair I-It". (Buber 1970, p. 53). It can be argued that the internal transformation of the "I" involves the dialectical sublation of elements in the ego that prevent the I–You relationship. However, according to Buber, this transformation involves (a divine) grace external to the "I". As such it differs from the dialectical formula according to Hegel.

19   In anthropological terms, dialectical relations lead to forms of *integration* and assimilation, in contrast to dialogical relations that leads to *pluralism*. Theo-dialogical thinking, in my opinion, demonstrates the best effort to answer one of the most significant ethical dilemmas from the Enlightenment to the present: how to foster plurality while preserving a coherent moral perspective and avoiding moral relativism. That is, how to maintain a moral viewpoint with a clear sense of purpose and direction as well as a practical understanding of redemption, while also accepting and allowing the inherent differences that exist between people and between cultural groups, and realizing that this pluralism is itself "the will of God".

20   Buber's criticism of the Halacha is well known; he did not believe in ritual or in fixed prayer. For him, religious rituals were not a direct encounter with the Eternal Thou (see Kaplan 1996, pp. 82–84). The idea that revelation, as the emergence of the moral law, originates from the actual encounter with the other (and that the law is not a product of historical-mythological revelation, or the product of reasoning alone) is central to the philosophy of Levinas: "The relation with the other—the absolutely other—who has no frontier with the same is not exposed to the allergy that afflicts the same in a totality, upon which the Hegelian dialectic rests. The other is not for reason a scandal which launches it into dialectical movement, but the first rational teaching, the condition for all teaching" Levinas (1979, p. 203).

21   It should be noted that there was a change in Buber's philosophy. Dialectical reasoning is present in his early writings, in the period before he developed his dialogical philosophy. For example, in his early article "Judaism and Mankind", he writes "The eternal is born out of contradiction" (Buber 1972, p. 23), and "A unity born out of one's own duality and the redemption from it. [. . .] It [Judaism] can only offer, ever anew, a unification of mankind's diverse contents, and ever new possibilities for synthesis". (Buber 1972, p. 32). A comparison between Buber's early and later philosophy in the context of dialectic deserves an attentive study, which goes beyond the scope of this paper.

22   It is worth noting a difference between Heschel and Buber's approaches to prayer. Although both are existential thinkers whose philosophies were greatly influenced by the biblical prophets, the Haggadic-Rabbinic literature, and especially Hassidism, their positions are nevertheless different. Heschel did not accept Buber's idea that revelation is realized only through human communication, and, generally, Heschel refuses to reduce transcendent (i.e., ineffable) reality to secular notions. Heschel develops

the philosophical idea of prayer in accordance with his view of the traditional observance of it. His approach to Halacha is closer to Franz Rosenzweig's than to Buber's.

23  Buber writes in *I and Thou*: "And even as prayer is not in time but time in prayer." (Buber 1970, p. 59). This illustrates that prayer is not an initiated liturgical event, but a spontaneous existential experience that expands beyond its "religious" concept. However, unlike psychological and anthropological approaches, Buber does not reduce God to a merely mental process. This is reflected in the similarity between prayer and sacrifice compared to the magical action: "In prayer man pours himself out, dependent without reservation, knowing that, incomprehensibly, he acts on God, albeit without exacting anything from God; for when he no longer covets anything for himself, he beholds his effective activity burning in the supreme flame. [. . .] Magic wants to be effective without entering into any relationship and performs its arts in the void, while sacrifice and prayer step "before the countenance", into the perfection of the sacred basic word that signifies reciprocity. They say You and listen" (Buber 1970, pp. 130–31). Horwitz claims that Buber's interest in prayer is peripheral. However, if we refer to prayer as a moment of an actual "I-Eternal Thou'Thou" dialogue, then prayer is central to Buber's philosophy, much more so than the picture presented by Horwitz indicates. See Horwitz (1999, p. 294).

24  Some see prayer as an act of worship, an expression of faith, or a way of receiving God's mercy. Others see prayer as an attempt (not necessarily dialogical) to influence God or to please him, similar to an act of sacrifice. Even as an expression of getting closer to God and at the same time distancing oneself from all the things that prevent getting closer to God, prayer is often seen as psychologically strengthening the believer and not necessarily as a dialogical situation.

25  Levinas considers prayer to be "the human act of blessing God in living a life for the Other". For Levinas, this does not replace traditional prayer. Rather, prayer conditions ethical life. Ephraim Meir demonstrates that praying for the nonsuffering of the I is valid if it is a prayer to God who suffers through the suffering of man. Indeed, prayer is not perceived by Levinas as an open dialogue with God, but it is, at the very least, a model for the desired moral relations between human beings (E. Meir 2004, p. 146).

26  An example of this can be found in the ethics of Emmanuel Levinas. The theological situation—being in a relationship with the transcendent God—is understood as being in relationship with the transcendent *other* human being. For Levinas, revelation (of God's word) is the moral command radiating from *the face* of the other. In this context, he explicitly declares that *epiphany* replaces dialectical inference (see, e.g., Levinas 1979, pp. 77–78, 194–96). Levinas presents a discussion in phenomenological terms that corresponds with Hegel's thought. Levinas accepts Hegel's argument against Kant that the exteriority of a being is inscribed in its essence (and does not contradict it, as Kant contends). But unlike Hegel, Levinas claims that this unity between exteriority and essence is not a (logical-dialectical) conclusion of logical-dialectical reasoning, "but the epiphany that occurs as a face" (Levinas 1979, p. 196).

27  The concept of God as the idea of the "good" and the guarantee for the future fulfillment of morality and the Messianic idea given by the biblical prophets as the a priori basis for moral socialism, are topics that Cohen deals with already in the early period when religion was included in the ethics of his system of philosophy.

28  According to Cohen, it is not that ethics and morality depend on religion but rather that moral consciousness and moral behavior embody religiosity. This wording could seemingly allow him to bypass the need to explain religious laws whose moral meaning is vague. Despite this, Cohen clarifies the meaning of some religious rituals, such as wearing tzitzit and phylacteries (tefillin). Cohen saw these religious ceremonies as a substitute for ancient sacrificial worship and as symbolic memories. That is, they have no sacred or mystical meaning in themselves. Cohen continues the rational line of Moshe Mendelssohn, according to which the Jewish religious rituals are a symbol whose purpose is to awaken moral consciousness (Cohen 1995, p. 394).

29  As an "original form", the meaning is that through a philosophical inquiry in the aesthetic-literary form of the concrete prayer, it is possible to comprehend monotheism. The question arises: is it possible to understand the prayer in its depth without the experience of praying? According to Rosenzweig, who saw himself as correcting weaknesses in Cohen's idealistic approach, the transition from cognition (Erkenntnis) to experience (Erlebnis) is necessary.

30  Cohen's discussion of prayer continues his discussion of the relationship between the religious commandment and the law of ethics. In ethics, claims Cohen, there is a constant tension between the "internal" aspect of the law (i.e., the categorical imperative) and the concrete external written laws. The religious commandment, if understood correctly, includes both sides without causing this tension. Cohen argues that the commandments, in general, and prayer, in particular, correct the problem of the abstractness of the categorical imperative: the commandments are present in every moment of daily life and linked to every daily action. In other words, one does not "pray" a categorical imperative in the morning, but rather prays the morning prayer.

31  See also BT, Berakhot 32a; Maimonides, *Mishneh Torah*, Prayer and the Priestly Blessing, 4: 15–16. The fact that religious activities are worthless unless they are performed with the intention of the heart is what distinguishes prayer from magic.

32  (Kant 2009, pp. 215–19): "Praying, conceived as an inward formal service of God and hence as a means of grace, is a superstitious delusion (a fetishism). For, it is a mere *declaration of wishing* [erklärtes Wünschen] directed toward a being that needs no declaration of the inward attitude of the person wishing; thus nothing is done through it and therefore none of the duties incumbent upon us as commands of God are performed, and hence God is actually not served. [. . .] words and formulas can at best carry with it only the value of a means for repeated invigoration of that attitude within ourselves, but it cannot directly have any reference to divine pleasure and precisely therefore also cannot be everyone's duty; for, a means can be prescribed only to one who requires it

for certain purposes; yet far from everyone has a need for this means (to speak within and properly with himself, but allegedly all the more comprehensibly with God), but one must rather, through continued purification and elevation of the moral attitude [. . .] for this purpose speech is only a means for the power of imagination".

33 The central Jewish prayer: "Hear, O Israel! The Lord is our God, the Lord alone" (Deuteronomy 6:4) indeed emphasizes that God is One, and there is agreement on this among monotheists. But this explanation is not sufficient. God's unity means primarily his transcendence (Cohen 1995, pp. 35, 41). God's Oneness excludes polytheism. That is, the multiplicity of gods and their desires do not allow for a unified concept of morality and humanity. Cohen's God idea involves the dialectical articulation of unity; namely, the God idea presents not a unit or one among many but a unique God, whose uniqueness corresponds to the oneness of inclusion of all people who collectively belong to humanity.

34 Cohen saw the pantheistic approach, especially that of Spinoza, as one that led to the formation of secular-naturalistic philosophies. Pantheism, according to Cohen, gives an illegitimate and metaphysical description of God and is unable to give an adequate philosophical account of ethics. In a pantheistic perspective, an anthropomorphic identification of God with nature eliminates the possibility of morality as based on the a priori of reason. In this view, there is no longer a distinction between God as *being* and the world as *becoming*; hence, a moral purpose and the autonomy of the moral will cannot be maintained. In other words, according to Cohen, when a pantheist asserts to have a concept of an "eternal good", he ceases to be a pantheist because he cannot draw the concept from the ever-changing world. In other words, pantheism collapses nature into the metaphysics of secularism. See (Bienenstock 2011; Nauen 1979, pp. 111–24) and compare with (Melamed 2018, pp. 171–80). Melamed refers only to Cohen's "*Spinoza on State and Religion: Judaism and Christianity*" and not to Cohen's attitude toward pantheism and Spinoza in *Religion of Reason*. Hence, Melamed does not deal with the problem of pantheism as a mediation problem, as it is in Christianity according to Cohen. Melamed claims that Cohen misidentified Spinoza as a pantheist instead of as a panentheist (a position in which transcendence is possible).

35 In Cohen's early philosophical system, God appears in the context of the "principle of origin", whereas in his late religious philosophy, God appears in the framework of "correlation". See (M. Meir 2003).

36 See also there in p. 212: "This nearness I gain in God's forgiveness. Sin alienates me from God; forgiveness brings me near again. And thus is formed an unceasing two-way communication between God and the human soul: the longing and the bliss, consisting in trust".

37 For Cohen, revelation is an a priori condition for reason, while for Hegel, revelation is a primitive symbolic form that will be replaced by a philosophical concept of the absolute spirit.

38 In the traditional Jewish approach, a distinction is usually made between individual prayer and public prayer. In terms of the latter, Cohen contends that an internal connection is created between the members of the community through the communal element of prayer. But for Cohen, this connection is not political, with the aim of creating national particularism, but rather sociological in that it serves as a common language for all people from different layers of the community: rich and poor, young and old, etc. However, the main part of his discussion of the ethical phenomenology of prayer revolves around individual prayer.

39 According to Cohen, the instruction to "Love the other (the neighbor, your fellow) as you love yourself: I am the Lord" (Leviticus 19, 18) is a concrete demand to acknowledge the individual in front of you for their uniqueness, and not only as a part of the all (Mehrheit). This acknowledgment changes the *Nebenmensch* (next man) to a *Mitmensch* (fellowman).

40 (Cohen 2021, pp. 38–39): "To be sure [. . .] a direct use of the dialectical method is in itself impressive, but it is also doubtless a source of error. However, it does not point to the actual reason for error. The error is located in the pantheistic core of the system. That is, it is pantheism that centers the system of philosophy and all Being in nature".

41 The messianic hope for the realization of ideal morality and the unity of mankind is a straightforward consequence of monotheism. Namely, in Cohen's approach, history must be viewed in the context of the messianic idea, as it is expressed in the prophets' universal socio-moral vision. Cohen stresses that prophetic monotheism is characterized by its rejection of mythology.

42 On this subject, Cohen expresses a deviation from the traditional approach, as it is, for example, presented by Maimonides, Cohen's revered teacher, who writes (in Mishneh Torah, Foreign Worship and Customs of the Nations 1:1) that monotheism preceded polytheism. In the 19th century, based on the assumption that such a complex idea as monotheism could not have developed in the human consciousness in one moment, so to speak "out of nowhere", it was common to see the development of monotheism as a slow gradual development in small steps. Funkenstein argues that contemporary religious studies have abandoned this evolutionist version (see Funkenstein 1997).

43 He writes: "Sacrifice is not controversial in prophetic thought alone: it is included in the law", and "prayer did not originate in a polemic but in pure messianic naivete" (Cohen 1995, p. 310).

44 Prayer, writes Cohen, includes the entire content of monotheistic worship. Hence, when the centrality is given to a ritual activity that is not prayer, this means that this religion moves away from being monotheistic. Alternatively, a wrong understanding of monotheism will lead to wrong worship, and eventually to an immoral view. Cohen demonstrates a certain degree of apologetics concerning Christianity (mainly Catholicism) of his time. The sources of Judaism teach that not only is Judaism a religion of reason, but it is also the origin of this idea.

45 Another dialectical component that remains from the ancient worship within the evolved worship (i.e., by means of sublation) is the group ritual. Although the development of the form of worship resulted in the consolidation of the status of the *individual*,

who does not need any additional mediation other than his own prayer in correlation with his God, nevertheless, the collective element remains: Cohen writes, "The individual cannot and does not want to exist without the congregation, and Messianism demands that the congregation should be extended to mankind" (Cohen 1995, p. 395). The concept of "mankind" in philosophical ethics is, however, criticized in Cohen's philosophical-religious approach. According to Cohen, we should reject the assumption that the concept of the congregation is dialectically replaced by the concept of mankind. This is consistent with the idea that the categorical imperative is not a substitute for prayer. Just as Hegel saw the importance of the German nation not being dialectically swallowed up in the concept of humanity, Cohen, in a similar way, saw the importance of the particularity of the Jewish nation, which is divided into congregations.

46  (Rosenzweig 1979, p. 791): "It is not just the categorical imperative, but also the categorical indicative, that is pagan" (my translation).

47  Leo Strauss thought that a weakness in Cohen's neo-Kantian philosophical system was precisely because it complemented Kant's ethics with the Hegelian premise of a necessary dialectical progress in history. See (Strauss 1997, p. 18).

48  Cohen's criticism was directed at philosophies that developed out of a departure from Kant—including Hegel, Nietzsche, and Marx—which indirectly also included a critique of secularism. Secularism, he believed, is subject to the risk of arbitrary non-pluralistic universalism and anthropocentric pride that disconnects itself from "created reason", which is fundamentally correlative. After all, the correction of Kant's ethics in Cohen focuses on the discovery of the individual in the general (*Mehrheit*) as essential to the universal. It should be noted that Cohen accepted Kant's version of dialectic as a means of identifying errors (the negation of conclusions based on incorrect premises) and avoiding the reliance of knowledge (i.e., knowledge about God) on the senses.

49  Mendes-Flohr writes, "He [Rosenzweig] has, as is well known, his roots in German philosophical idealism. Although he later developed serious misgiving about this school of philosophy, he remained indebted to its tendency to see the history of thought and culture as dialectically interrelated and unified. And so he sees Judaism". (Mendes-Flohr 1992, p. 190).

50  Unlike Hegel, Rosenzweig believes that dialectical synthesis is not produced by the union of two ends but rather by two ends that define and alter themselves with one another (e.g., belief and knowledge both redefine themselves with respect to one another). However, complete unification is not attained. (Rachel-Freund 1979, p. 95).

51  See also Rosenzweig (2005, p. 314). Instead of a combination of philosophy and theology, there is a combination of the philosopher and theologian: "The philosopher must be more than philosophy [. . .] he must pray the prayer of creatures [. . .]. And the theologian must be more than theology. [. . .] He must be truthful; he must love God [. . .] he must say the prayer".

52  It should be noted that philosophy for Rosenzweig itself is defined, under the influence of Schelling, as re-mythological narrating praxis. (Rosenzweig 2000, pp. 35–36).

53  Through dialogue with his convert friend Eugen Rosenstock, who was a keen intellectual, scientist, and at the same time devout Christian, Rosenzweig realized that religious belief does not have to contradict a scientific position on the world and history (see Glatzer 1961, pp. xiv–xv). Rosenzweig's decision not to convert to Christianity and to dedicate himself to Jewish life occurred as "an inner call"; he heard God addressing him by his first name. Hearing and answering this call is, in his eyes, an event of "rebirth" (see Horwitz 1981, p. 30).

54  According to Rosenzweig, Creation is God's first revelation (i.e., God's relation with the world). Hence, the notion of creation is based on the notion of revelation; however, there is no direct chronological continuation from the former to the latter. Creation "does not develop dialectically" from revelation, writes Rosenzweig; rather, it is its "inversion".

55  Rosenzweig describes the Jewish holidays and their prayers according to this key. For example, the Sabbath is a creation holiday; the festivals of revelation are Pesach (Passover), Shavuot, and Sukkot; whereas the holidays of redemption are Yom Kippur and Rosh Hashanah (the Days of Awe: The New Year and the Day of Atonement) (Rosenzweig 2005, pp. 330–46).

56  This position prevails in the tradition; see the illuminating essay of (Peli 1973).

57  "Death" in Rosenzweig's *The Star,* as a starting point for his criticisms of idealist philosophy in general and Hegel in particular, has been extensively discussed. The Star begins with the statement: "From Death, it is from the fear of death that all cognition of all the All begins" (Rosenzweig 2005, p. 9). See also, for example, (Gibbs 1992, pp. 36–40; P. E. Gordon 2003, pp. 165–74; Dagan 2000, 2001). Rosenzweig opens *The Star* with a critique of Hegelian idealism, which, in his opinion, failed to deal with the fear of death by offering man the eternity of reason. Hegel, according to Rosenzweig, sees the positive meaning of death (that is, erasing the boundaries of the subject) in the dialectical process of negation, whereas Rosenzweig sees this as a negative and terrifying matter for which remedy and redemption must be offered. In his approach, proposing abstract matters as a substitute for the concrete, accessible, and tangible is not philosophically satisfactory. Hence, Rosenzweig believes that God must be revealed—a God that has expression in the inner, immediate, existential experience.

58  Rosenzweig eliminates the dialectical relations between Judaism and Christianity as they appeared in the Hegelian approach, according to which Christianity is a dialectical replacement of Judaism. According to Rosenzweig, Christianity stands alongside Judaism as the two parts of the truth.

59  In *Genesis Rabbah* 11:5, it is said that the river Sambation rested on the Sabbath. The residents saw this as a sign from God and therefore kept the Sabbath laws. Rosenzweig writes, "If, instead of the Main, it was this river [Sambation] that flowed through Frankfurt, there is no doubt that the whole Jewish community there would strictly observe the Sabbath. But God does not give

such signs. [...] God obviously wants only those who are free for his own. [...] So he has no choice: he must tempt man [...] And on the other hand, man must also reckon with this possibility that God only "tempts" him [...] Therefore the mutual possibilities of tempting meet in prayer, that of God and that of man; prayer is harnessed between these two possibilities; while being afraid of being tempted by God, it yet knows in itself the power of tempting God himself" (Rosenzweig 2005, pp. 284–85).

[60] In this context, we should note that, similar to other thinkers of his time (Hermann Cohen, Ernst Simon, and Martin Buber), Rosenzweig indeed distinguishes between prayer that characterizes monotheism and the magic that characterizes idolatry. Generally, in magic, a human being tries to force God to do as he wishes. The success of this attempt means turning God into an idol and religion into idolatry (see Rosenberg 1996, p. 93; Pinkas 2021).

[61] (Buber 1970, p. 131): "What distinguishes sacrifice and prayer from all magic? Magic wants to be effective without entering into any relationship and performs its arts in the void, while sacrifice and prayer step "before the countenance", into the perfection of the sacred basic word that signifies reciprocity. They say You and listen".

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
