# Peer review of "On Prayer and Dialectic in Modern Jewish Philosophy: Hermann Cohen and Franz Rosenzweig"

_religions, doi:10.3390/rel14080996_

Round 1
Reviewer 1 Report
Line 41 - "many" seems too vague here. You do have a footnote that gives several examples, but that does not constitute "many." Perhaps use a phrase like "significant scholarly trends".
Line 218 - "cannot be denied" seems too strong. Maybe soften the language a little.
Line 242 - "the most explicit" Again, seems too absolute. Soften a little to "among the most explicit" or something
Line 323 - the isolated sentence is a little awkward without being attached to a paragraph
Line 331 - another isolated sentence unattached to a paragraph. Is this a norm of which I am unaware?
Line 358 - "is known" by whom? "THE great" - you mean the ONLY great? Is there a consensus on him being THE great neo-Kantian Jewish philosopher? If so, among whom is Cohen known as such? Your argument will be more powerful throughout if you are more specific in your claims rather than more sweeping and broad in those claims.
Line 371 - MUCH better. This is a great example of a non-over-reaching claim. More of this needed throughout.
The paper is excellent and I recommend it for publication with a few caveats. First, there is no abstract. Looks like you left the template's abstract in there instead of your own. You'll have to add one to give the reader a clearer idea of the purpose of your paper. Second, we have to wait until line 316 until you clearly articulate the purpose of your paper. While you are laying important groundwork in order to get to that point, it is probably advisable for you to get to a clear statement of the paper's purpose a little earlier than that.
The isolated sentences unattached to paragraphs throughout are out of place and jarring. I advise making them part of the paragraphs that precede or follow them.
Some of the prose in your paper is more over-reaching than I am used to in academic writing. Phrases like "THE most important philosopher" or "it is impossible to argue against ..." These sorts of phrases could be softened to make the same point but avoid the problem of making them so far-reaching as to be practically indefensible.
Some of the sections drag on a little. You have organize the paper into clearly defined sections, but I wonder if you'd be open to placing some subsections within them so that the reader doesn't get lost? You have one such extensive research in all of your footnotes that I think having subsection headings would help the reader who is inevitably going to peruse those footnotes follow the logical flow of your argument more easily
After making these changes, I am ready to recommend this paper for publication. Well done.
Author Response
Response to Reviewer 1,
Many thanks for your careful reading and helpful comments.
I have addressed all comments: Please see the revised version: I marked some of the changes in yellow.
- The main point (lines: 41, 218, 242 and more) was the softening of the language.
- Line 358 I removed "the great".
- I removed the isolated quotes (line 323, 331).
- An abstract has been added.
- I have moved many of the footnotes to the main text, and others to the end, so that reading the article will be more fluent and easy. I deleted unclear comments or changed the wording to make it clearer.
- I also added a list of sources at the end according to the publishing requirements.
Regarding adding subsections: I could add subsections on Buber, Heschel, and perhaps Levinas, but their ideas are used mainly as examples, and I don't want to create the impression that what is said about them encompasses their positions.
This article is the result of the teaching of a course on this topic, and I hope that it will contribute to researchers and teaching staff in the field (hence the many footnotes to research literature, especially in the first parts).
I thank you very much for the comments that helped to improve the scientific validity of the claims in this study.
With Kind regards,

Reviewer 2 Report
I enjoyed reading your paper. It deals with a crucial moment of modern German-Jewish speculation and the comparative analysis of Cohen, Buber and Rosenzweig proves itself as highly enlightening. My only suggestion would be, regarding the comparison between Rosenzweig´s and Cohen´s different philosophical attitudes. In the context of prayer and revelation this seems to be expressed also in their different Jewish medieval ideal figures, i.e. Cohen's turn to Maimonides's "ethics" as against Rosenzweig´s turn toward Judah Halevi´s poetics (and liturgical Piut in general).
Here are some specific remarks:
1. The copy I received for review entails no abstract and no list of references. I´m not sure about the necessity f the latter but certainly, you need to add an abstract.
2. While the paper is well written and well organized some lines are corrupted in the present PDF file I reviewed (note lines 382, 389)
3. Fn. 29: "Maimonides on Genesis...". Did you mean Nachmanides? Maimonides has no pntateuch commentary. In this case, the reference must e to one of his works.
4. Fn. 69 - full reference to Georg Kohler´s paper. It was already brought in a short reference before (see Fn. 61).
5. "(Rosenzweig) defined The Star as “a system of philosophy” and not a philosophy of religion." (lines 763-4). I agree. But philosophy itself s defined, together with the late Schelling and the famous fragment published by Rosenzweig a few years earlier, as re-mythological narrating praxis.
The language quality is good. technical problems in the present PDF file,
Author Response
Dear reviewer,
Many thanks for your careful reading, helpful comments, and positive review.
I addressed all the comments and recommendations. Please see the attached file for the updated version (please note, that the markings in yellow refer to the changes made according to the evaluators' comments).
- I added the comment about Rambam and Rabbi Yehuda Halevi in the closing remarks. In the attached version, the changes are marked in yellow (see line 1044). This topic is important, which is why I addressed it in the concluding remarks and not in a footnote.
- I added an abstract and a list of references.
- I have removed isolated quotes at the beginning of the sections (382, 389, and more) that created the technical problem in the PDF file.
- Special thanks for noticing the mistake in the comment about Maimonides. It is indeed Nachmanides. See fn. 15.
- I have arranged all the footnotes according to the publishing requirements. And this refers to correcting abbreviations in repeated footnotes.
- Regarding Rosenzweig and Schelling and the philosophical method. I have addressed your comment, please see fn. 52.
Many thanks for the careful reading and helpful comments, which improved this article.
With kind regards.
Sincerely,
